# An order-to-disorder structural switch activates the FoxM1 transcription factor

Aimee H Marceau[1†], Caileen M Brison[1†], Santrupti Nerli[1,2†], Heather E Arsenault[3], Andrew C McShan[1], Eefei Chen[1], Hsiau-Wei Lee[1], Jennifer A Benanti[3], Nikolaos G Sgourakis[1], Seth M Rubin[1]*

[1]Department of Chemistry and Biochemistry, University of California, Santa Cruz, Santa Cruz, United States; [2]Department of Computer Science, University of California, Santa Cruz, Santa Cruz, United States; [3]Department of Molecular, Cell and Cancer Biology, University of Massachusetts Medical School, Worcester, United States

**Abstract** Intrinsically disordered transcription factor transactivation domains (TADs) function through structural plasticity, adopting ordered conformations when bound to transcriptional co-regulators. Many transcription factors contain a negative regulatory domain (NRD) that suppresses recruitment of transcriptional machinery through autoregulation of the TAD. We report the solution structure of an autoinhibited NRD-TAD complex within FoxM1, a critical activator of mitotic gene expression. We observe that while both the FoxM1 NRD and TAD are primarily intrinsically disordered domains, they associate and adopt a structured conformation. We identify how Plk1 and Cdk kinases cooperate to phosphorylate FoxM1, which releases the TAD into a disordered conformation that then associates with the TAZ2 or KIX domains of the transcriptional co-activator CBP. Our results support a mechanism of FoxM1 regulation in which the TAD undergoes switching between disordered and different ordered structures.
DOI: https://doi.org/10.7554/eLife.46131.001

**\*For correspondence:**
srubin@ucsc.edu

[†]These authors contributed equally to this work

**Competing interests:** The authors declare that no competing interests exist.

## Introduction

The transactivation domains (TADs) of transcription factors are responsible for recruiting transcriptional machinery to target promoters (*Ptashne and Gann, 1997*). TADs consist of structurally disordered domains, which facilitate adaptable association with multiple protein partners and, in some contexts, form interaction network hubs within liquid condensates (*Cho et al., 2018*; *Chong et al., 2018*; *Liu et al., 2006*; *Minezaki et al., 2006*; *Wright and Dyson, 2015*). Considering their intrinsic disorder, it is critical to understand how TAD accessibility is restricted to regulate transcription factor activity. Accessibility can be modulated through cellular localization, co-factor binding, ligand binding, posttranslational modifications, protein stability, phase separation, or oligomerization (*Bah et al., 2015*; *Cho et al., 2018*; *Chong et al., 2018*; *Wright and Dyson, 2015*). Many transcription factors, including critical regulators of cell division, contain a negative regulatory domain (NRD) that is thought to bind and inhibit the TAD (*Kim et al., 1999*; *Park et al., 2008b*; *Ramsay and Gonda, 2008*; *Shi et al., 1995*; *Spengler and Brattain, 2006*; *Wierstra and Alves, 2006a*). Little is understood about how the NRD blocks accessibility of the intrinsically disordered TAD and how TAD release is achieved. Here we have identified the structural mechanisms underlying regulation of the mammalian Forkhead box M1 transcription factor (FoxM1).

FoxM1 is critical for dividing cells, as it activates expression of genes that facilitate mitotic entry, the mitotic program, and proper cell-cycle progression (*Fu et al., 2008*; *Korver et al., 1997*; *Korver et al., 1998*; *Laoukili et al., 2005*; *Wang et al., 2005*; *Wonsey and Follettie, 2005*). FoxM1 expression is normally confined to dividing cells and is found most commonly in embryogenesis,

hematopoiesis, and tissue repair (*Korver et al., 1997*; *Ramakrishna et al., 2007*; *Ustiyan et al., 2009*). FoxM1 is essential for proper development; null mice exhibit an embryonic lethal phenotype due to liver, blood, heart, and lung abnormalities resulting from proliferation defects (*Ramakrishna et al., 2007*).

Aberrant expression and misregulation of FoxM1 are associated with multiple cancers and are directly related to increased proliferation, metastasis, chemoresistance, and poor therapeutic prognosis (*Koo et al., 2012*; *Myatt and Lam, 2007*; *Raychaudhuri and Park, 2011*). FoxM1 misregulation at both the transcriptional and protein level has been observed in adenocarcinomas, breast cancer, squamous cell carcinomas, leukemia, lymphoma, and many additional malignancies (*Kwok et al., 2010*; *Millour et al., 2010*; *Park et al., 2011*; *Raychaudhuri and Park, 2011*; *Xu et al., 2013*; *Yang et al., 2013*). Several studies have shown that FoxM1 activates tumor metastasis, mediates drug resistance, and regulates pluripotency-associated genes responsible for maintaining tumor cells in their undifferentiated state (*Carr et al., 2010*; *Kwok et al., 2010*; *Millour et al., 2010*; *Park et al., 2011*; *Raychaudhuri and Park, 2011*; *Wang et al., 2011*; *Weigelt et al., 2005*; *Xie et al., 2010*). Additionally, deletion of FoxM1 in cancer cells inhibits tumor development and growth and leads to increased apoptosis (*Kalinichenko et al., 2004*; *Wonsey and Follettie, 2005*). Chemical inhibition of FoxM1 has become an important therapeutic strategy that would benefit from further definition of FoxM1 structure and the biochemical mechanisms that control activation (*Gormally et al., 2014*; *Radhakrishnan et al., 2006*).

Structural characterization of transcription factors remains challenging as they are enriched in low complexity or intrinsically disordered domains (*Babu et al., 2011*). Transcription factors may have folded domains, such as a DNA binding domain (DBD), but these domains are typically modular and interspersed among disordered sequences. FoxM1 contains four distinct functional domains (*Figure 1A*): a negative regulatory domain (NRD), a DNA binding domain (DBD), a region containing several cyclin-dependent kinase (Cdk) consensus sites (called here the Cdk site region or CSR) and a TAD (*Anders et al., 2011*; *Fu et al., 2008*; *Littler et al., 2010*; *Major et al., 2004*; *Park et al., 2008b*; *Wierstra and Alves, 2006a*; *Wierstra and Alves, 2006b*). The DBD is the only region with a known well-ordered structure; it binds to promoter sequences of many cell-cycle regulated genes (*Littler et al., 2010*). The NRD suppresses FoxM1 activity in transcription reporter assays, and it has been proposed that the NRD binds and sequesters the TAD from its association with the co-activators CBP and p300 (*Laoukili et al., 2008a*; *Major et al., 2004*; *Park et al., 2008b*; *Wierstra and Alves, 2006a*). Phosphorylation by either Cdks or polo-like kinase 1 (Plk1) inhibits repressive activity of the NRD in these assays (*Anders et al., 2011*; *Fu et al., 2008*; *Laoukili et al., 2008a*; *Major et al., 2004*; *Park et al., 2008b*), but it is not clear how this activation step occurs.

Here we determine the structural mechanisms underlying FoxM1 repression and activation, providing a picture for how transcription factor autoregulation occurs. Using solution NMR together with biophysical and cellular assays, we determine a structural model for the autoinhibited conformation, identify the key Cdk and Plk1 phosphorylation events that disrupt the NRD-TAD association, and characterize TAD association with the KIX and TAZ2 domains of the transcriptional co-activator protein CBP. Remarkably, we find that the key phosphorylation-induced activation step is marked by a transition from structural order to disorder, which results in accessibility of the TAD for co-activator binding.

## Results

### The FoxM1 NRD directly associates with the TAD

The FoxM1 NRD inhibits transcription factor transactivation activity early in the cell cycle. Transcription reporter assays suggest that the NRD negatively regulates the TAD through a sequestration mechanism (*Park et al., 2008b*; *Wierstra and Alves, 2006a*). To confirm a direct interdomain association and to define more precisely the NRD and TAD domain boundaries, we purified recombinant protein constructs of various lengths and assayed their association in trans using isothermal titration calorimetry (ITC) (*Figure 1*). We first used an NRD-containing construct that includes the entire N-terminus of FoxM1 up until near the start of the DBD (residues 1–203 in human isoform FoxM1b) and a long C-terminal construct (residues 526–748) including the Cdk-site rich region (CSR) and putative TAD. We observe binding between these protein fragments and measured an affinity of

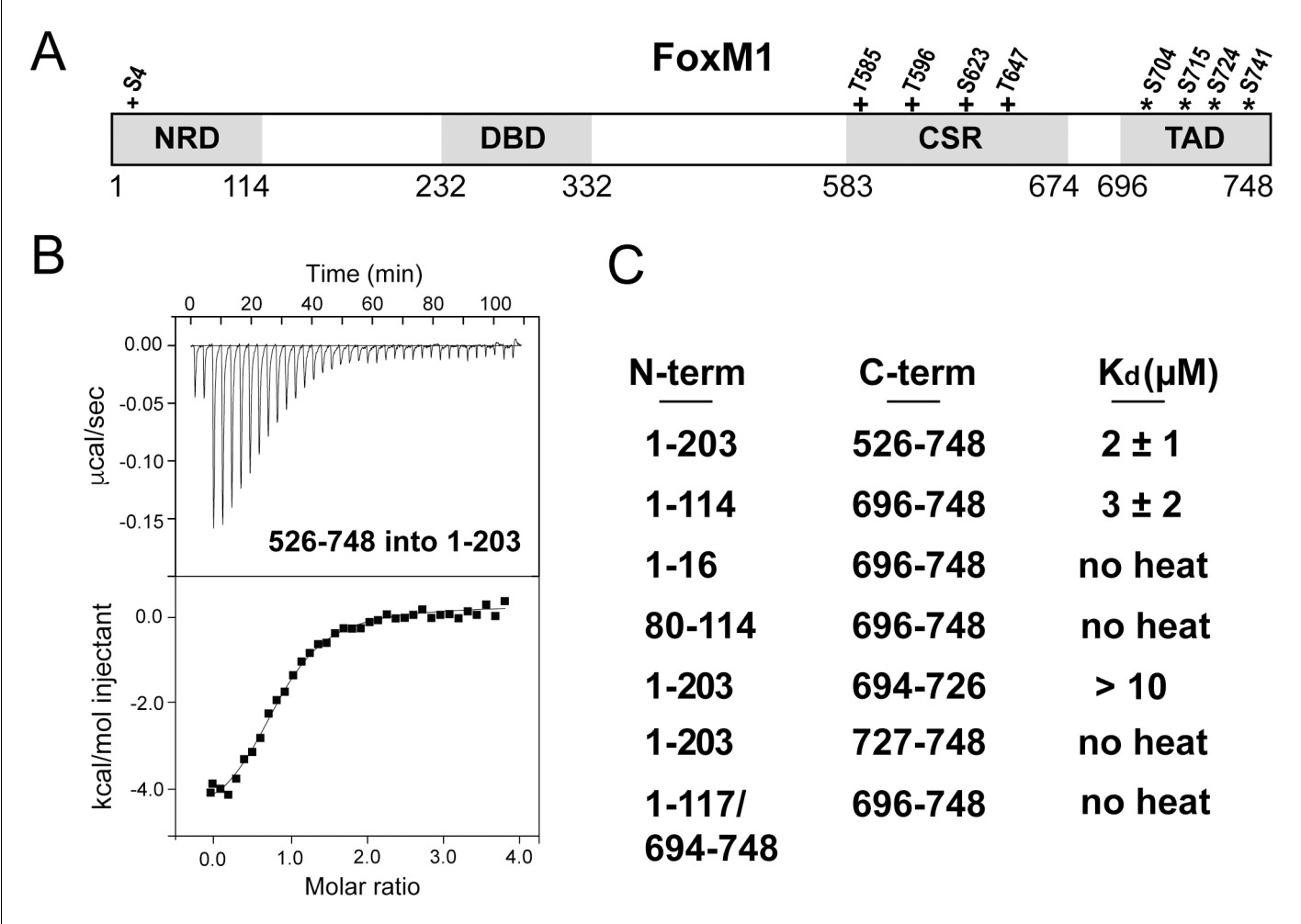

**Figure 1.** Direct association between the FoxM1 NRD and TAD. (**A**) Domain boundaries for the negative regulatory domain (NRD), DNA binding domain (DBD), Cdk-site region (CSR), and transactivation domain (TAD). Numbering for human FoxM1 isoform b is used. Conserved consensus Cdk (+) phosphorylation sites and the Plk1 phosphorylation sites (*) identified here are indicated. (**B**) Isothermal titration calorimetry (ITC) data indicate association between purified TAD and NRD-containing constructs. (**C**) ITC affinity measurements for the indicated purified protein constructs. From these data, we define the NRD and TAD boundaries in panel A. All values from the ITC data fitting are listed in *Supplementary file 1*.
DOI: https://doi.org/10.7554/eLife.46131.002

$K_d = 2 \pm 1$ µM (*Figure 1B and C*). Using sequence conservation we designed a minimal NRD (1-114) and minimal TAD (696-748) and found that they associate with comparable affinity ($K_d = 3 \pm 2$ µM). When we attempted to shorten the constructs further, we found loss of affinity (*Figure 1C*), so we conclude that these fragments contain the approximate sequences necessary and sufficient for inter-domain association.

## NMR structure of the NRD-TAD interface

We combined NMR with Rosetta structural modeling to understand how the FoxM1 NRD binds the TAD in the autoinhibited conformation (*Nerli and Sgourakis, 2019*). NMR studies of TAD-coactivator complexes have primarily utilized an approach in which the two domains are purified separately with isotope labeling of only one protein (*De Guzman et al., 2006*; *Radhakrishnan et al., 1997*; *Zor et al., 2004*). We found that spectra of the human NRD-TAD complex assembled from independently purified domains were generally of poor quality. We alternatively purified a protein construct for structural studies in which the minimal NRD and TAD were fused by a short cleavable linker (1-

117/694-748). This fusion protein does not bind additional TAD in trans (*Figure 1C*), which suggests that the TAD and NRD are associated within the fusion.

We observed optimal NMR spectra using the protein sequence from zebrafish, deleting an internal loop in the NRD (residues 21–41 in the zebrafish sequence, *Figure 2A*), cleaving the fusion linker, and uniformly deuterating the nonexchangeable hydrogen positions (*Figure 2—figure supplement 1* and *Figure 2—figure supplement 2*). We assigned 64% of the backbone $H^N$, $N^H$, $C\alpha$, and $C\beta$ chemical shifts in this construct using triple-resonance backbone correlation spectra with TROSY readout. Using TALOS-N (*Shen and Bax, 2015*; *Shen et al., 2009*), we calculated the secondary structure index (SSI) and estimated backbone order parameter (RCI-$S^2$, 0 is total disorder and 1 is fully rigid) for the assigned regions of both domains (*Figure 2B and C*). We find significant structural order within a set of residues including sequences from both the NRD and TAD. Importantly, within this ordered region, the backbone chemical shifts for 71 out of 75 non-proline residues were assigned. Notably, the SSI analysis suggests that a stretch of residues in the TAD adopts a β-hairpin conformation, which contrasts the helical TAD structures typically observed in complexes with co-activator domains (*De Guzman et al., 2006*; *Goto et al., 2002*; *Krois et al., 2016*; *Radhakrishnan et al., 1997*; *Wang et al., 2013*; *Zor et al., 2004*).

To build a model of the core NRD-TAD structure, we prepared perdeuterated protein samples with isoleucine, leucine, and valine (ILV) methyl sidechain $^{13}C/^1H$ labels. We performed a combination of isotopomer-selective TOCSY experiments and several NOESY experiments to assign and validate the resonances of ILV methyls and to generate a set of NOE-based distance restraints for structure calculations (see Materials and methods) (*Otten et al., 2010*). We then combined the chemical shifts, NOE restraints, and RDCs to generate a set of structures using customized RASREC-Rosetta calculations, as described in Materials and methods (*Supplementary file 2*). We included only the core, structured sequences of each domain in the calculation (*Figure 2A and C*). Sequences outside the core region are likely disordered, as suggested by no observable NOEs and low order parameters for assigned residues in the flanking regions. The structural ensemble of the NRD-TAD complex, which consists of ten top scoring models based on Rosetta energy function and good overall fit to the experimental data, showed high convergence with heavy atom root mean square displacements of 0.48 Å for core residues (*Figure 2D*). The observed backbone conformation in these calculated models match TALOS-N chemical shift predictions of secondary structure (*Figure 2B*) and fits experimental RDC data well (*Figure 2—figure supplement 3*). The core structured elements in the NRD and TAD correlate well with the sequences of highest conservation in those domains, and nearly all the residues that are key for forming the interface are conserved (*Figure 2A*).

The NRD-TAD structure consists of a five-stranded pleated β-sheet and a single α-helix. Three of the beta strands (β1-β3) are from sequences in the NRD, and two additional strands are from sequences in the TAD (β4-β5) (*Figure 2E and F*). The amphipathic helix is an insertion between the parallel β2 and β3 strands in the NRD. One face of the helix packs against the β-sheet to form the hydrophobic core of the structure. Sidechains from β−strands in both the NRD (I64, V76, I78, F106, and L108, human sequence numbering) and TAD (L709 and L716) form an extended, buried interface through highly specific, experimentally confirmed interactions with helix residues (I84, I87, I88, and L91) (*Figure 3A*). The fact that both the NRD and TAD contribute essential residues to the hydrophobic core suggests the requirement of an association for forming the observed structure of both domains. We explore this idea further below.

The structured region of the TAD consists of a 12 amino acid sequence that adopts a β-hairpin conformation and binds the NRD by extending the β-sheet. The NRD-TAD interface therefore consists of interstrand backbone hydrogen bonds between NRD β3 and TAD β4 and several van der Waals contacts between sidechains in those β−strands and the NRD helix (*Figures 2F* and *3A*). The backbone hydrogen-bonding network is supported by unambiguously assigned interstrand amide-amide NOE cross-peaks (*Figure 2F*), and, likewise, the sidechain interactions are supported by methyl-methyl NOEs (*Figure 3A*). Sidechain contacts at the interface are observed between I88, L91, F106, I107, and L108 in the NRD and L703, V708, L709, and L716 in the TAD (*Figure 3A*). Interestingly, even when interacting with the NRD,~20% of the TAD is structurally well defined, while the majority of the TAD remains intrinsically disordered. We propose that this small, structurally plastic region of the TAD is key for the regulation of FoxM1 activity.

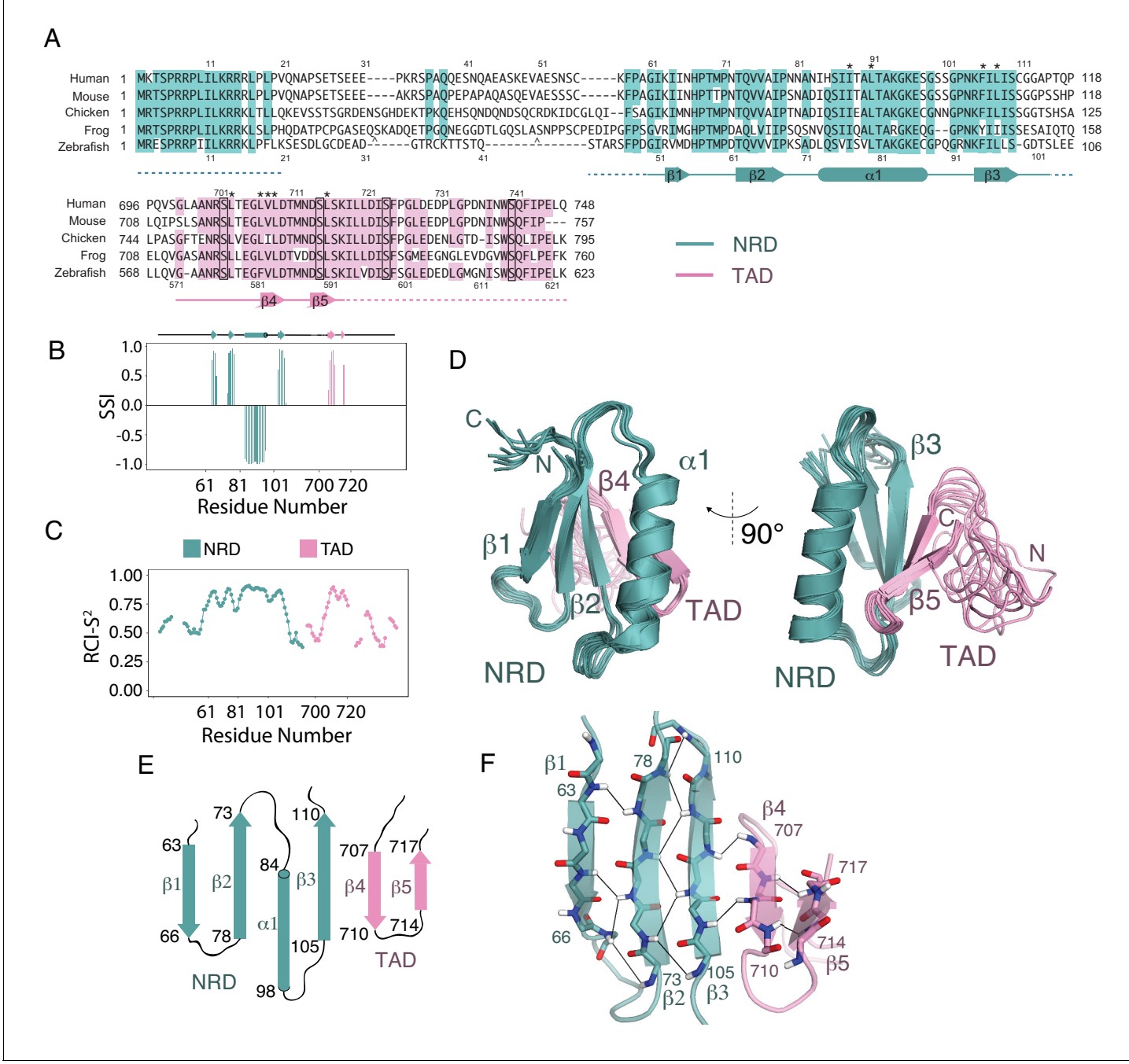

**Figure 2.** Solution structure of an NRD-TAD complex. (**A**) Sequence alignment of FoxM1 orthologs from *Homo sapiens* (human), *Mus musculus* (mouse), *Gallus gallus* (chicken), *Xenopus laevis* (frog), and *Danio rerio* (zebrafish). Conserved residues in at least four of the five sequences are colored. Secondary structure assignments are determined from dictionary of protein secondary structure (DSSP) analysis of the final NMR ensemble (***Kabsch and Sander, 1983***). Dashed lines indicate residues that are present in the NMR construct but are not included in the structure calculations and are significantly disordered according to the backbone chemical shifts. The asterisks (*) mark residues for which interdomain (NRD-TAD) NOEs have been unambiguously assigned. The carets (ˆ) mark poorly conserved sequence insertions not shown in the frog sequence. Plk1 phosphorylation sites in the TAD are boxed. (**B**) Secondary Structure Index (SSI) derived from TALOS-N analysis of backbone chemical shifts corresponding to residues in the NRD (cyan) and TAD (pink) domains. Positive SSI values are consistent with β-strand and negative values are consistent with α helical structure. Amino acid numbering corresponding to the human sequence is used. (**C**) Estimated backbone order parameters (Random Coil Index RCI-$S^2$) derived from the chemical shifts are shown for residues in the NRD and TAD (***Berjanskii and Wishart, 2005***). Lower RCI-$S^2$ values indicate flexibility, higher RCI-$S^2$ values indicate rigidity. (**D**) Overlay of ten final Rosetta models guided by the chemical shift, NOE, and RDC data. (**E**) Topology diagram of the NRD and TAD domains. (**F**) Structure of the five-stranded β-sheet. Unambiguously assigned interstrand amide proton-proton NOEs are shown as lines.

*Figure 2 continued on next page*

*Figure 2 continued*

DOI: https://doi.org/10.7554/eLife.46131.003

The following figure supplements are available for figure 2:

**Figure supplement 1.** Details of the zebrafish FoxM1 fusion construct used for NMR analysis.

DOI: https://doi.org/10.7554/eLife.46131.004

**Figure supplement 2.** Comparison of NMR data between the NRD-TAD fusion construct with the linker cleaved and uncleaved.

DOI: https://doi.org/10.7554/eLife.46131.005

**Figure supplement 3.** Correlation between the experimental and predicted RDC measurements of the NRD-TAD structural ensemble.

DOI: https://doi.org/10.7554/eLife.46131.006

## Disruption of the NRD-TAD interaction activates FoxM1

To probe the structural role of the NRD-TAD interface in repressing FoxM1 transactivation, we first tested the effects of mutations in the NRD and TAD on the interdomain association using the ITC assay with human FoxM1 domains (*Figure 3B*). For these experiments, we used the TAD (696-748) and a longer NRD (1-203), which were the most stable constructs containing each domain in solution and the simplest to purify. We expressed and purified NRD constructs (residues 1–203) containing mutations in each of the three β-strands. These mutations, which replace hydrophobic sidechains for alanine, were made to residues on either the face of the β-sheet that interacts with the helix (proximal) or the face pointing away from the helix (distal). We find that mutations on the β-sheet face that is proximal to the helix all inhibit binding with the TAD, with mutations on the two strands closest to the TAD (V76A/I78A in β2 and F106A/L108A in β3) having the greatest effects. We find that mutations to the hydrophobic face of the NRD helix (I88A and L91A) also disrupt TAD binding. In contrast, mutations to the distal side of the NRD β-sheet only result in a modest decrease in affinity (I65A or V75A) or no change in affinity (I107A/I109A). We also substituted alanines for sidechains in the TAD near the NRD interface. We found that three mutations have strong effects on binding (V708A, L709A, L716A), while mutation at one sidechain that points toward the distal side of the sheet (M712A) has a modest effect.

We next tested the effects of mutations to the NRD-TAD interface on FoxM1 activity in U2OS osteosarcoma cells using a luciferase reporter assay (*Figure 3C* and *Figure 3—figure supplement 1*). We expressed WT and mutant FoxM1 together with a plasmid that contains six tandem Forkhead-response elements (6DB) or the promoter of PLK1 (PLK1p), a FoxM1 target gene, upstream from the luciferase gene. As previously described, luciferase activity increases upon expression of WT FoxM1 compared to transfection of empty vector, which reflects the ability of FoxM1 to transactivate luciferase expression from the promoters in the reporter plasmid (*Anders et al., 2011*; *Laoukili et al., 2005*). We found that this activity is further increased upon expression of FoxM1 harboring mutations that destabilize the NRD-TAD interface. The cell reporter data generally correlate well with our in vitro ITC binding data. For example, a β1 strand mutant (I62A/I64A), which only produces a modest loss of binding affinity, does not significantly change FoxM1 activity in the cell assay. In contrast, mutations closer to the interface in NRD strands β2 and β3, the NRD helix, and the TAD have strong effects. We conclude that the WT FoxM1 activity in this assay is mitigated by autorepression and that, by destabilizing the NRD-TAD structure and association, these mutations inhibit autorepression and result in higher activity. One interesting exception to the observed correlation between the biochemical and cellular assays is the L716A mutation. This mutation results in complete loss of affinity but has a more modest activating effect in the luciferase assay compared to other TAD mutations. We explore below an additional role for this residue in activation through recruitment of CBP/p300 co-activator.

## Phosphorylation of Cdk consensus sites in FoxM1 does not modulate NRD-TAD affinity

In the current accepted model for FoxM1 activation, Cdk phosphorylation relieves NRD inhibition by directly destabilizing the NRD-TAD association (*Anders et al., 2011*; *Laoukili et al., 2008a*; *Park et al., 2008b*; *Wierstra and Alves, 2006b*). Deletion of the NRD in cells results in constitutively active FoxM1 independent of Cdk activity, and adding back of the NRD in trans restores the autoinhibition. Previous work has highlighted the importance of specific Cdk sites; for example, alanine

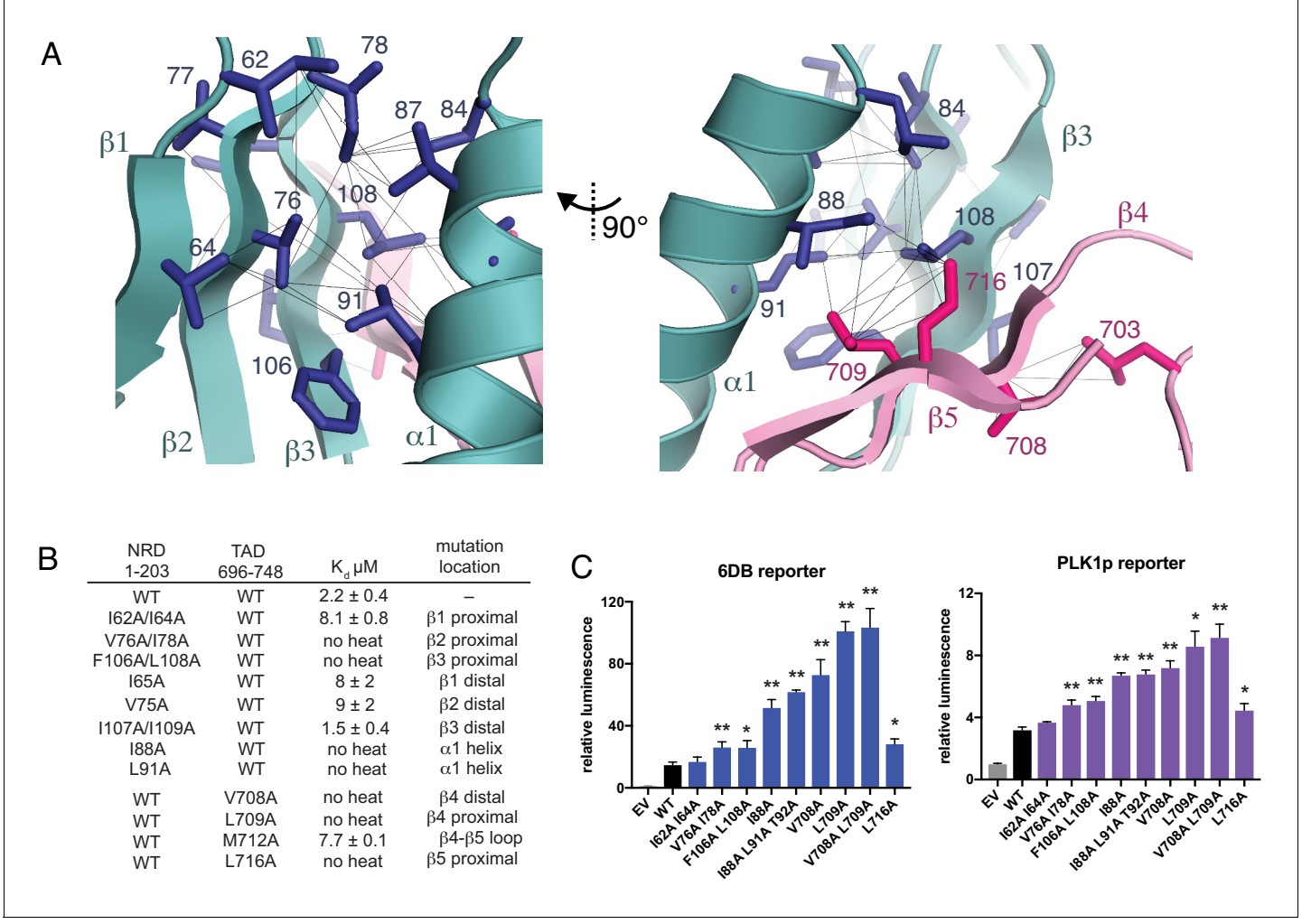

**Figure 3.** Interactions stabilizing the NRD-TAD hydrophobic core and interface. (**A**) Hydrophobic sidechains forming the structural core of the NRD-TAD complex. Human amino acid numbering is indicated. Unambiguously assigned ILV methyl-methyl NOEs are shown as lines. (**B**) ITC measurements of the NRD-TAD binding affinity using the indicated WT or mutant domains. Mutations were chosen on the side of the β-sheet that is either proximal or distal to the α-helix. (**C**) Luciferase reporter assays of FoxM1 transactivation activity. Reporter plasmids containing the luciferase gene downstream of either six repeats of a FoxM1 responsive element (6DB, left) or the PLK1 promoter sequence (PLK1p, right) were co-transfected with WT or mutant FoxM1 into U2OS cells. EV is empty vector. Significant differences in the relative luminescence from WT are indicated with asterisks: *p<0.05, **p<0.01 (using two-tailed student's t-test). For expression and cell cycle controls, see *Figure 3—figure supplement 1*.

DOI: https://doi.org/10.7554/eLife.46131.007

The following figure supplement is available for figure 3:

**Figure supplement 1.** Data supporting luciferase transcription reporter assay.

DOI: https://doi.org/10.7554/eLife.46131.008

mutations of T585, T596, S623, and S678 result in decreased FoxM1 activation (*Anders et al., 2011*; *Fu et al., 2008*; *Lüscher-Firzlaff et al., 2006*; *Major et al., 2004*). However, other results suggest a cumulative effect of Cdk phosphorylation activity on FoxM1 activation (*Anders et al., 2011*; *Lüscher-Firzlaff et al., 2006*; *Wierstra and Alves, 2006b*). We addressed with purified domains whether phosphorylation of Cdk sites in human FoxM1 directly affects the inhibitory NRD-TAD association. For these assays, we phosphorylated Cdk consensus sites using Cdk2-Cyclin A (Cdk2-CycA) as previously described (*McGrath et al., 2017*), we verified phosphorylation by electrospray mass spectrometry (*Figure 4—figure supplement 1*), and we measured affinities using the ITC assay. We first observed that phosphorylation of the NRD, which contains a single conserved consensus Cdk site (S4), has minimal effect on the affinity of the NRD for the CSR-TAD (*Figure 4A*). While there are

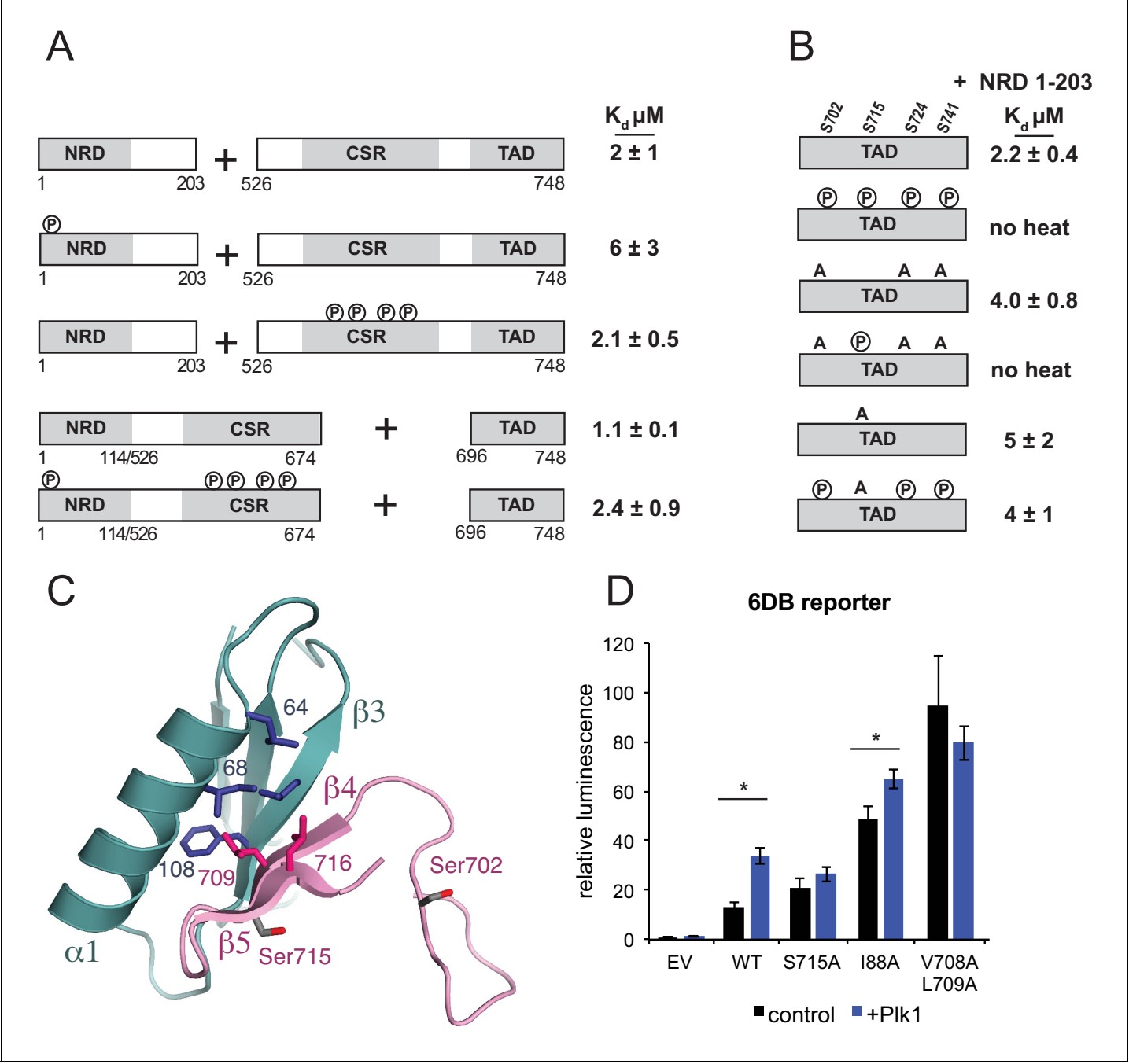

**Figure 4.** Plk1 phosphorylation of Ser715 inhibits NRD-TAD association. (**A**) ITC binding affinities of NRD and TAD-containing FoxM1 fragments following Cdk phosphorylation as indicated. (**B**) ITC binding affinity of NRD (1–203) for WT TAD (696-748) and TAD containing mutations at Plk1 sites. Measurements were made with and without Plk1 phosphorylation. (**C**) Ser715 is near the interface in the NMR structure of the NRD-TAD complex. Human amino acid numbering is used. (**D**) Luciferase reporter expression from the 6DB promoter as in *Figure 3C*. Only experiments in which significant differences in the relative luminescence between expression of FoxM1 alone (control, black) or co-expression with Plk1 (purple) are indicated with asterisks (*$p<0.05$, using two-tailed student's t-test).

DOI: https://doi.org/10.7554/eLife.46131.009

The following figure supplements are available for figure 4:

**Figure supplement 1.** Electrospray mass spectrometry characterization of kinase reactions.
DOI: https://doi.org/10.7554/eLife.46131.010

**Figure supplement 2.** Phosphorylation dependent CSR association with the Plk1 polobox domain.
DOI: https://doi.org/10.7554/eLife.46131.011

no consensus Cdk sites in the TAD, there are several sites in the Cdk site region (CSR) (**Figure 1A**). To test if Cdk phosphorylation of this region directly affects the NRD-TAD association, we generated protein constructs in which the CSR is included with the TAD or fused to the NRD. The CSR-TAD construct (amino acids 526–748) binds the NRD with similar affinity whether or not it is first phosphorylated with Cdk2-CycA (**Figure 4A**). The NRD-CSR fusion protein (amino acids 1–114 + 526–674) binds the TAD with similar affinity as the NRD alone and with similar affinity whether or not it is first phosphorylated with Cdk2-CycA. These measurements are consistent with the lack of Cdk sites near the NRD-TAD interface in our structural model. The observation here that phosphorylation of Cdk consensus sites has no direct effect on the repressive NRD-TAD interaction suggests that Cdk phosphorylation is not sufficient for activation of the FoxM1 transcription factor.

## Plk1 phosphorylation of the TAD inhibits NRD association to activate FoxM1-dependent gene expression

Plk1 activity has also been implicated in FoxM1 activation (**Fu et al., 2008**). The FoxM1 TAD contains five potential serine Plk1 phosphorylation sites. We phosphorylated the FoxM1 TAD by incubating with purified Plk1, and we confirmed quantitative phosphorylation of 4 sites (Ser702, Ser715, Ser724, and Ser741) with electrospray mass spectrometry (**Figure 4—figure supplement 1**). These sites are all conserved (**Figure 2A**), and two of them (Ser715 and Ser724) were previously identified as Plk1 sites in cells (**Fu et al., 2008**). We observed that the Plk1 phosphorylated-TAD no longer binds the NRD by ITC (**Figure 4B**). Of the four Plk1 sites, Ser715 is the only site that appears near the structured NRD-TAD interface (**Figure 4C**). Using TAD constructs with serine to alanine mutations, which cannot be phosphorylated, we tested the importance of specifically phosphorylating Ser715 on the NRD-TAD affinity. When the three other Plk1 sites are mutated, the phosphorylated TAD still lacks any detectable affinity for the NRD. In contrast, phosphorylation of a Ser715A construct results in no change in affinity. We conclude that Ser715 phosphorylation by Plk1 is necessary and sufficient for inhibiting the NRD-TAD association. These results are consistent with and explain the previous report that an Ser715A/Ser724A mutation decreases FoxM1 activation of several mitotic genes in U2OS cells (**Fu et al., 2008**). Ser715 is in strand β5 near the NRD-TAD interface; however, the sidechain is exposed to solvent, and it is not certain that phosphorylation directly inhibits the association through electrostatic repulsion. We suggest that phosphorylation destabilizes the β-hairpin conformation and/or stabilizes an alternative conformation of the TAD (**Andrew et al., 2002**; **Riemen and Waters, 2009**).

We used the luciferase reporter assay to probe how Plk1 phosphorylation of Ser715 influences autoinhibition in U2OS cells (**Figure 4D**). Similar to as previously described (**Fu et al., 2008**), we find that expression of Plk1 along with WT FoxM1 increases the 6DB-reporter expression 2.5-fold. No significant increase in reporter expression is observed if Plk1 is expressed with a FoxM1 S715A mutant, which is consistent with our in vitro data demonstrating that S715 phosphorylation is necessary for TAD-NRD inhibition. As above, we again observe that mutations to the NRD (I88A) or TAD (V708A/L709A) result in increased expression of reporter relative to WT. However, we only observe a slight 1.3-fold additional increase in reporter expression when Plk1 is co-expressed with FoxM1 I88A and no significant increase with FoxM1 V708A/L709A. We conclude that because the NRD-TAD interface is already destabilized in these mutants, phosphorylation does not result in additional activity. These results support our model that Plk1 phosphorylation of Ser715 activates FoxM1 by dissociating the repressive NRD-TAD interface.

Our results show that Plk1 phosphorylation and not Cdk directly inhibits the NRD-TAD association, yet strong evidence points to Cdk phosphorylation as an important step in the activation of FoxM1 during the cell cycle (**Anders et al., 2011**; **Fu et al., 2008**; **Laoukili et al., 2008a**; **Lüscher-Firzlaff et al., 2006**; **Major et al., 2004**; **Park et al., 2008b**; **Wierstra and Alves, 2006a**; **Wierstra and Alves, 2006b**). In the CSR of FoxM1, there are two Cdk sites that when phosphorylated create canonical binding sites for the Plk1 polobox domain. These sites have been shown to recruit Plk1 to FoxM1 in co-immunoprecipitation experiments (**Fu et al., 2008**). We also found using purified proteins that Cdk phosphorylation of the CSR induces Plk1 association (**Figure 4—figure supplement 2**), confirming that the Plk1-FoxM1 association is direct and phosphorylation dependent. These results suggest that Cdk phosphorylation primes the polobox binding sites and enhances Plk1 binding and phosphorylation of the TAD. We propose that phosphorylation of the TAD by

Plk1 ultimately inhibits the autorepressive NRD-TAD interaction, freeing the TAD to bind to co-activator CBP.

## The dissociated FoxM1 NRD and TAD domains are largely unstructured

To probe the structures of the NRD and TAD once they are dissociated, we acquired circular dichroism (CD) spectra on isolated human NRD (residues 1–114) and TAD (694-748) domains and compared them to the human homolog of the NRD-TAD fusion construct used for NMR (1-117/694-748; Δ25–50) (*Figure 5A* and *Figure 5—figure supplement 1A*). We used the minimal NRD construct for CD analysis, because NMR data suggest the region between 115–203 is disordered (*Figure 5—figure supplement 2*). Consistent with the NMR model, the spectrum of the fusion reflects the

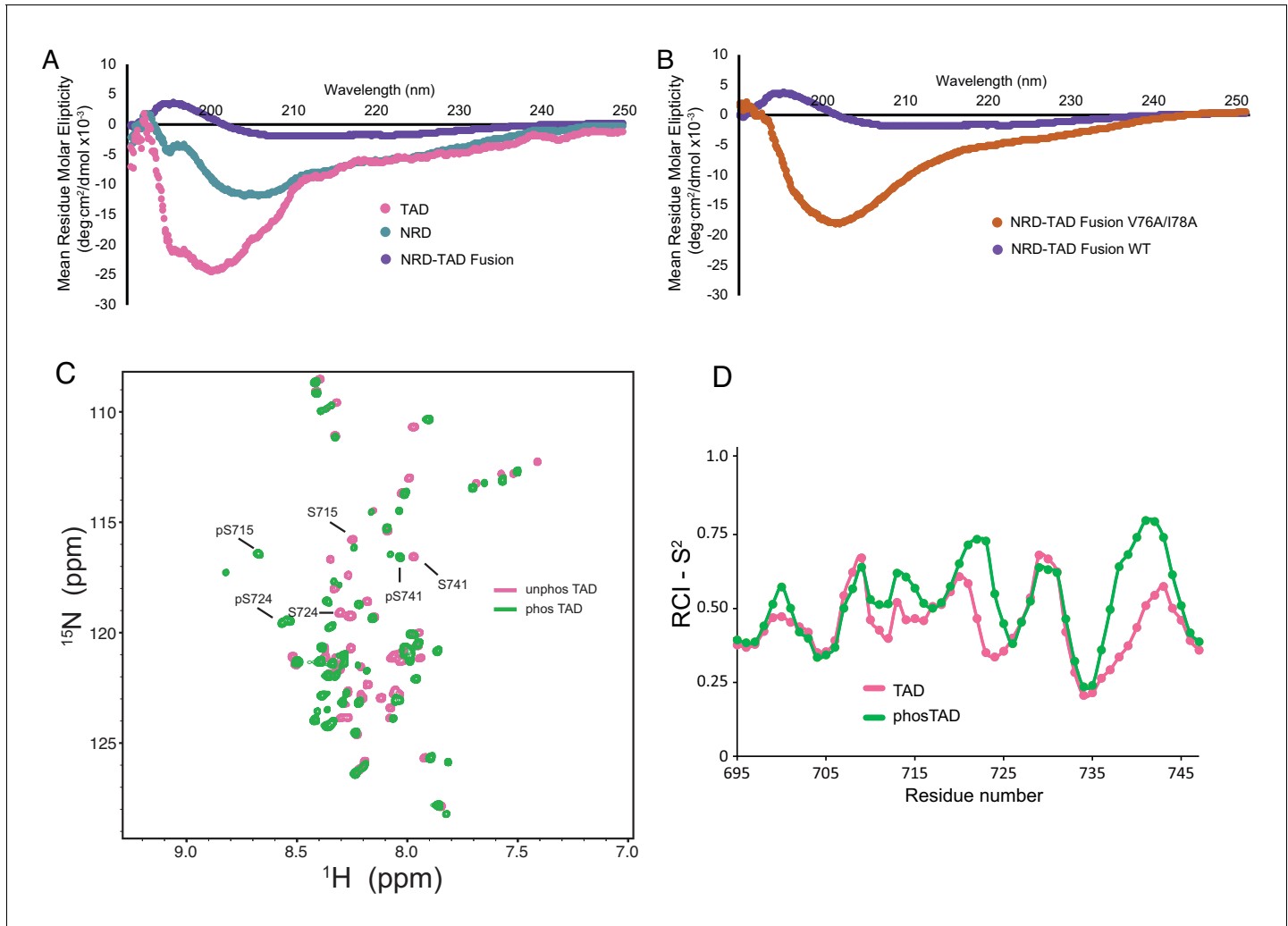

**Figure 5.** The dissociated NRD and TAD domains are intrinsically disordered. (**A, B**) CD spectra of the indicated purified proteins. (**C**) 2D $^1$H-$^{15}$N HSQC spectra of the zebrafish FoxM1 TAD (corresponding to residues 696–748 in the human protein) either unphosphorylated (pink) or phosphorylated (green) recorded at 25°C at 800 MHz. Sequence specific assignments (examples shown for three Plk1-phosphorylated serines) were made using backbone triple resonance correlation spectra. (**D**) RCI-S$^2$ order parameter plot of assigned residues in the free TAD and Plk1-phosphorylated TAD.

DOI: https://doi.org/10.7554/eLife.46131.012

The following figure supplements are available for figure 5:

**Figure supplement 1.** Additional circular dichroism measurements reporting the structural order of the associated NRD-TAD complex and disorder of the isolated TAD and NRD domains.
DOI: https://doi.org/10.7554/eLife.46131.013

**Figure supplement 2.** Additional data supporting the conclusion that the isolated NRD is highly disordered.
DOI: https://doi.org/10.7554/eLife.46131.014

presence of structure. In contrast, the TAD and NRD spectra indicate relatively greater disorder, although they do reflect some secondary structure content. Because we could not efficiently phosphorylate the fusion in vitro with Plk1, we mimicked the dissociative effect of phosphorylation by introducing the V76A/I78A mutation. Like phosphorylation, this interface mutation inhibits NRD-TAD association in the ITC assay (*Figure 3B*). The CD spectrum of the V76A/I78A fusion protein also reflects significant disorder (*Figure 5B*), which supports the conclusion that NRD-TAD binding and structure formation are interdependent. This interdependence is further supported by the observation that the structural core of the NRD-TAD complex is formed by hydrophobic residues from both domains. Notably, introduction of the same destabilizing V76/I78A mutation in the NRD alone has modest effects on the CD spectrum (*Figure 5—figure supplement 1B*), which is consistent with the conclusion that the WT NRD alone is already considerably disordered.

To further probe the structure of the TAD alone and the effects of Plk1 phosphorylation, we collected NMR data and assigned the backbone amide chemical shifts for the zebrafish FoxM1 TAD (residues 571–623, which corresponds to human 696–748), using both unphosphorylated and Plk1-phosphorylated samples (*Figure 5C*). 47 out of 50 of the non-Pro residues in the TAD and phosphorylated TAD were assigned. Using TALOS-N as above, we calculated the RCI-S$^2$ order parameter for residues in both the phosphorylated and unphosphorylated TAD (*Figure 5D*). The data support the conclusion that both the phosphorylated and unphosphorylated isolated TAD are primarily unstructured. These results in comparison with the NMR data for the NRD-TAD complex (*Figure 2B–2D*) suggest that the β-hairpin structure of the TAD is adopted upon binding to the NRD.

The CD spectra of the NRD and the dissociated mutant NRD-TAD fusion also reflect significant structural disorder (*Figure 5A and B*). Despite varying sample conditions and using orthologous sequences from several organisms, NMR HSQC spectra of the isolated NRD were of poor quality, likely reflecting a tendency to aggregate, but are also consistent with significant disorder within the domain (*Figure 5—figure supplement 2A and B*). In further support of the conclusion that the NRD is poorly structured when not in complex with the TAD, we found that the NRD alone elutes through a size exclusion column as expected for a protein lacking a globular structure (*Figure 5—figure supplement 2C*). The observations that the NRD and TAD appear unstructured when alone suggest that the repressive NRD-TAD association drives the domains to fold into the inhibitory conformation, and conversely that activation of FoxM1 is marked by an order-to-disorder structural transition.

## FoxM1 TAD binds the KIX and TAZ2 domains of CBP only when released from the NRD

The requirement of Cdk phosphorylation for FoxM1 activity has been linked to FoxM1 recruitment of the co-activator acetyltransferase proteins CBP and p300 (*Major et al., 2004*). These co-activators share high sequence homology and contain several protein-protein interaction domains that associate with transcription factor TADs (*Wang et al., 2013*). We tested three common TAD interaction domains in CBP and found that purified TAZ2 and KIX domains bind the FoxM1 TAD, while TAZ1 does not (*Figure 6A*). We further studied the properties of FoxM1 binding to TAZ2. We could not detect binding of the NRD-TAD fusion protein to the TAZ2 domain using ITC, and we observed weak association to an NRD-TAD complex using NMR (*Figure 6—figure supplement 1*). The fusion protein containing the I88A interface mutation did bind in the ITC assay, albeit with slightly weaker affinity than the TAD alone (*Figure 6B*). These results support the model that the NRD inhibits FoxM1 activity by sequestering the TAD from interacting with transcriptional co-activators. Phosphorylation of the CSR-TAD by either Cdk or Plk1 does not affect the affinity with TAZ2 (*Figure 6C*). We conclude that TAD phosphorylation does not modulate its affinity for CBP by changing direct binding interactions. Instead our results indicate that phosphorylation shifts the NRD-TAD binding equilibrium toward TAD release from the NRD so that the TAD is accessible.

The structures of several TAD sequences bound to CBP or p300 KIX and TAZ2 have been determined, and it is found that the intrinsically disordered TADs invariably adopt helical conformations when associated with co-activator (*De Guzman et al., 2006*; *Goto et al., 2002*; *Krois et al., 2016*; *Radhakrishnan et al., 1997*; *Wang et al., 2013*; *Zor et al., 2004*). Binding typically entails contacts from hydrophobic residues along the face of one or more short amphipathic helices in the TAD. Motifs such as ΦXXΦΦ or ΦΦXXΦ (Φ is a bulky hydrophobic residue and X is any residue) are commonly observed in the TAD interacting sequences. Two sequences in the FoxM1 TAD contain such a motif and appear amenable to forming amphipathic helices (*Figure 6—figure supplement 2*). We

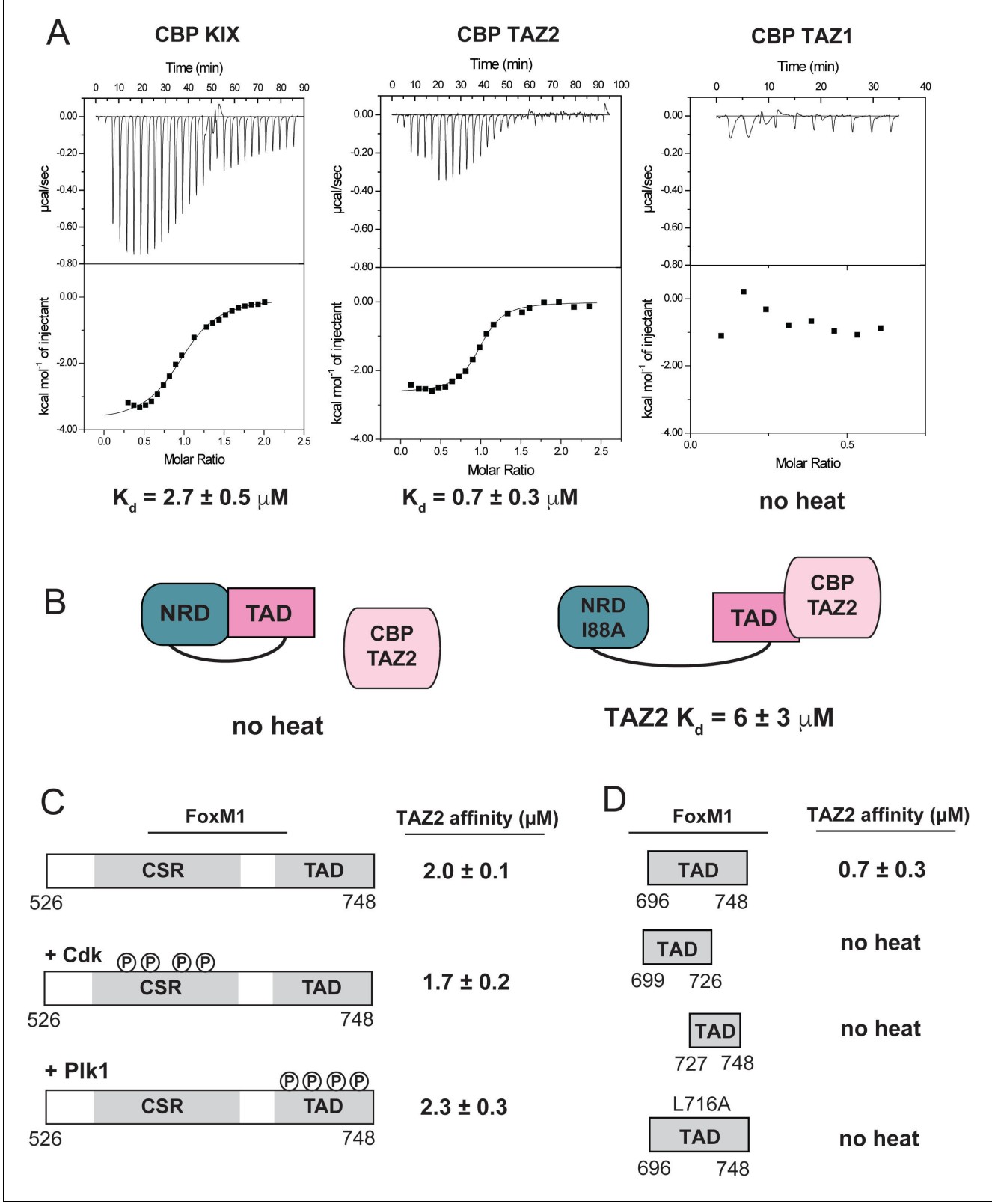

**Figure 6.** Direct binding of NRD-released TAD to CBP/p300. (**A**) Raw ITC data and the calculated affinities of FoxM1 TAD (696-748) binding to the KIX, TAZ1, and TAZ2 domains of CBP. (**B**) Affinities of WT and I88A NRD-TAD fusion for TAZ2. (**C and D**) ITC affinities of the indicated FoxM1 construct for TAZ2. Where indicated, the purified CSR-TAD domain was phosphorylated with Plk1 or Cdk2-CycA prior to the affinity measurement. All values from the ITC data fitting are listed in *Supplementary file 1*.

*Figure 6 continued on next page*

*Figure 6 continued*

DOI: https://doi.org/10.7554/eLife.46131.015

The following figure supplements are available for figure 6:

**Figure supplement 1.** NMR data indicating weak association between CBP TAZ2 and the cleaved NRD-TAD fusion.

DOI: https://doi.org/10.7554/eLife.46131.016

**Figure supplement 2.** Predicted short amphipathic helices in the TAD.

DOI: https://doi.org/10.7554/eLife.46131.017

---

find that both of these sequences are required for TAZ2 binding (*Figure 6D*). Notably, the predicted binding sequence around 713–726 overlaps with the sequence that forms the β-hairpin in binding the NRD in the repressed conformation. We tested a L716A TAD mutant and found no detectable binding to TAZ2 (*Figure 6D*). The importance of L716 for CBP association explains how NRD binding inhibits CBP association and likely explains why the L716A mutation did not result in similar increased FoxM1 activity in the luciferase assay as other mutations in the TAD (*Figure 3C*). The mutation disrupts both NRD inhibition and activation through CBP recruitment.

Overall, our results support a new structural model of FoxM1 regulation (*Figure 7*). In its autoinhibited conformation, the NRD binds the TAD via a structured interface and stabilizes a β-hairpin conformation that is incompatible with co-activator binding. Cdk phosphorylation of the CSR domain induces Plk1 docking to FoxM1 and subsequent phosphorylation of Ser715. Phosphorylation destabilizes the TAD β-hairpin, which releases from the NRD, and both the NRD and TAD become disordered. The TAD is free to adopt the helical conformation that is likely necessary for recruiting co-activator CBP through association with either the KIX or TAZ2 domain.

## Discussion

Eukaryotic transcription factors almost invariably contain large regions of intrinsic structural disorder, which facilitate protein-protein interactions for gene activation, posttranslational modifications for regulation, and control over nuclear localization (*Minezaki et al., 2006*; *Wright and Dyson, 2015*). Whereas structural disorder is required for function, it follows that the induction of structure could be used to negatively regulate transcription factor activity by restricting accessibility, particularly of the transactivation domain. Indeed, our results show that the NRD of FoxM1 is able to bind the TAD and induce a structured conformation that is incompatible with its binding to co-activator protein. Interestingly, the repressed TAD has a β-hairpin structure, while the CBP bound state likely contains α-helical structures and involves a TAD sequence near the Ser715 phosphorylation site that overlaps with the NRD-binding sequence. Thus, FoxM1 regulation entails secondary structure switching of the TAD between strand, coil, and helical structures.

We find that Ser715 phosphorylation destabilizes the NRD-TAD association and that the dissociated TAD is intrinsically disordered and capable of binding CBP. Phosphorylation has been observed to control structural transitions as a means to regulate other intrinsically disordered proteins. For example, both the transcription factor Ets-1 and the translation regulator 4E-BP2 undergo disorder-to-order transitions upon phosphorylation, while the nucleophosmin protein dissociates from a structured pentamer to a disordered monomer upon phosphorylation (*Bah et al., 2015*; *Mitrea et al., 2014*; *Pufall et al., 2005*). FoxM1 demonstrates a new example in which the order-to-disorder transition occurs coincident with modulation of an intramolecular association between domains. This plasticity of disordered domains and the ability to regulate the plasticity through posttranslational modifications explains why intrinsically disordered proteins are so well suited to regulate biological function.

Unexpectedly, our data indicate that in the active FoxM1 conformation, in which phosphorylation inhibits the repressive NRD-TAD association, the NRD is largely disordered as well as the TAD. We propose that this additional disorder may be important to increase conformational flexibility within the context of chromatin, to promote regulation of FoxM1 stability, or to contribute to the formation of low complexity domain condensates as recently described (*Cho et al., 2018*; *Chong et al., 2018*; *Laoukili et al., 2008b*; *Liu et al., 2006*; *Minezaki et al., 2006*; *Park et al., 2008a*; *Wang et al., 2017*; *Wright and Dyson, 2015*). While several transcription factor NRDs have been characterized (*Kim et al., 1999*; *Park et al., 2008b*; *Ramsay and Gonda, 2008*; *Shi et al., 1995*; *Spengler and*

*Brattain, 2006*; *Wierstra and Alves, 2006a*), the idea that NRDs may appear in regions of intrinsic disorder should motivate the identification of additional 'hidden' NRDs that adopt structure only upon interaction with their target.

Like many cell division regulatory proteins, FoxM1 undergoes cell-cycle dependent multisite phosphorylation. While multiple Cdk and Plk1 phosphorylation events have been implicated in FoxM1 activation, our results demonstrate that phosphorylation of one specific Plk1 site (Ser715) is directly responsible for freeing the TAD from sequestration by the NRD. This result clarifies an indirect role for Cdk in driving FoxM1 activity through priming the CSR for Plk1 docking and further phosphorylation of the TAD (*Fu et al., 2008*). The idea that different phosphorylation events have specific and distinct roles is in contrast to several proposed models for multisite phosphorylation in the cell cycle, in which a threshold aggregate of redundant phosphorylation events drives a change in protein activity (*Harvey et al., 2005*; *Kim and Ferrell, 2007*; *Nash et al., 2001*). Our observation of specificity in the effects of FoxM1 phosphorylation more resembles what has been proposed for the retinoblastoma protein, in which specific phosphorylation events drive distinct structural changes or protein-protein interactions (*Rubin, 2013*).

The structural picture revealed here of FoxM1 repression provides novel mechanistic insights into transcription factor regulation and may motivate novel cancer therapeutics. It has been concluded that FoxM1 inhibition would significantly impact our treatment of several cancers that express high levels of the protein (*Koo et al., 2012*; *Myatt and Lam, 2007*; *Raychaudhuri and Park, 2011*). However, consistent with the challenges of targeting transcription factors with chemotherapeutics, only few reports have described potential candidate molecules (*Gormally et al., 2014*; *Radhakrishnan et al., 2006*). The identification of a structured, repressed conformation suggests the possibility of developing molecules that bind and stabilize the repressive NRD-TAD association or target FoxM1 for degradation.

# Materials and methods

**Key resources table**

| Reagent type (species) or resource | Designation | Source or reference | Identifiers | Additional information |
|---|---|---|---|---|
| Cell line (human) | U2OS | other | RRID:CVCL_0042 | Paul Kaufman (UMass Medical School) |
| Antibody | anti-FoxM1, rabbit polyclonal | Bethyl | cat #: A301-533A-M | (1:2000) |
| Antibody | anti-actin, mouse monoclonal | Sigma | cat #: A1978 | (1:100,000) |
| Antibody | anti-Flag, mouse monoclonal | Sigma | cat #: F1804 | (1:500) |
| Recombinant DNA reagent | PGEX 4 T-3 | Addgene | | Engineered to contain TEV protease site. |
| Recombinant DNA reagent | pFastBac HTB | Addgene | | |
| Recombinant DNA reagent | pET-14B | Addgene | | Engineered to contain TEV protease site |
| Recombinant DNA reagent | pRcCMV myc-Plk1 | Addgene; PMID: 7962193 | | Erich Nigg |
| Recombinant DNA reagent | pGL3-6DB | PMID: 22094256 | | Peter Sicinski (Dana Farber Cancer Institute) |
| Recombinant DNA reagent | pGL3-PLK1 | PMID: 22094256 | | Peter Sicinski (Dana Farber Cancer Institute) |
| Recombinant DNA reagent | pCDNA3-FOXM1C | PMID: 22094256 | | Peter Sicinski (Dana Farber Cancer Institute) |
| Recombinant DNA reagent | pCDNA3-FOXM1C-I62A I64A | this paper | | mutation cloned into pCDNA3-FOXM1C |

*Continued on next page*

*Continued*

| Reagent type (species) or resource | Designation | Source or reference | Identifiers | Additional information |
|---|---|---|---|---|
| Recombinant DNA reagent | pCDNA3-FOXM1C-V76A I78A | this paper | | mutation cloned into pCDNA3-FOXM1C |
| Recombinant DNA reagent | pCDNA3-FOXM1C-I88A | this paper | | mutation cloned into pCDNA3-FOXM1C |
| Recombinant DNA reagent | pCDNA3-FOXM1C -I88A L91A T92A | this paper | | mutation cloned into pCDNA3-FOXM1C |
| Recombinant DNA reagent | pCDNA3-FOXM1C -F106A L108A | this paper | | mutation cloned into pCDNA3-FOXM1C |
| Recombinant DNA reagent | pCDNA3-FOXM1C-V723A | this paper | | mutation cloned into pCDNA3-FOXM1C |
| Recombinant DNA reagent | pCDNA3-FOXM1C-L724A | this paper | | mutation cloned into pCDNA3-FOXM1C |
| Recombinant DNA reagent | pCDNA3-FOXM1C -V723A L724A | this paper | | mutation cloned into pCDNA3-FOXM1C |
| Recombinant DNA reagent | pCDNA3-FOXM1C-L731A | this paper | | mutation cloned into pCDNA3-FOXM1C |
| Recombinant DNA reagent | pCDNA3-FOXM1C-S730A | this paper | | mutation cloned into pCDNA3-FOXM1C |
| Recombinant DNA reagent | pCDNA3 | Invitrogen | | |
| Recombinant DNA reagent | pCMV-Renilla | this paper | | Renilla luciferase cloned into CMV promoter expression vector |
| Software, algorithm | RASREC-Rosetta for structure calculations | https://csrosetta.chemistry.ucsc.edu | | |

## Protein expression

The human FoxM1 constructs (1–114, 1–203, 1–332, 80–114, 1-117/694-748 fusion, 1-117/694-748 fusion Δ25–50, 1-114/526-674 fusion, 573–635, 526–748, and 696–748) were expressed in and purified from *E. coli* as N-terminal GST fusion proteins with TEV cleavage sites. Cells were induced in mid-log phase with 1 mM IPTG, and cells were grown for 3–4 hr at 25°C. All proteins were purified from lysates with glutathione sepharose affinity chromatography. Fusion and C-terminal (TAD) constructs were further purified with Q-sepharose and then cleaved with TEV protease. Proteins were buffer exchanged into 25 mM Tris, 1 M NaCl, 5 mM DTT (pH 8.0) and then passed over glutathione sepharose resin to remove free GST, concentrated, and run over Superdex-75 (GE Healthcare) into 20 mM HEPES, 150 mM NaCl, 5% glycerol (pH 7.5). After TEV cleavage, N-terminal (NRD) constructs were diluted in 20 mM Hepes, 5 mM DTT (pH 7.0), and purified with S-sepharose and Superdex-75 as described for C-terminal constructs.

Human Plk1 kinase domain (13-345) was expressed with an N-terminal His tag in Sf9 cells using the FastBac expression system. Cells were harvested and lysed in a buffer containing 50 mM Tris, 300 mM KCl, 2 mM $MgCl_2$, 10 mM Imidazole, Sigma Protease Inhibitor (P8340), 2 mM phenylmethyl-sulfonyl fluoride (pH 8.0). Protein was purified with Nickel Sepharose Excel resin (GE Healthcare) equilibrated in lysis buffer. The resin was washed with a buffer containing 1 M KCl, 50 mM Tris, 40 mM Imidazole, 10 mM BME, 2 mM $MgCl_2$ (pH 8.0) and protein was eluted in 200 mM KCl, 50 mM Tris, 300 mM Imidazole, 10% glycerol v/v, 10 mM BME, 2 mM $MgCl_2$ (pH 8.0). Protein was dialyzed into storage buffer (200 mM KCl, 50 mM Tris, 10 mM BME, 2 mM $MgCl_2$, 10% glycerol v/v (pH 8.0)) and stored at −80 °C. The polobox domain of human Plk1 and Cdk2-CycA were expressed and purified as previously described (*Cheng et al., 2003*; *McGrath et al., 2013*).

*Mus musculus* CBP KIX (CBP residues 567–653), TAZ2 (residues 1764–1855) and TAZ1 (residues 340–439) were expressed in *E. coli*. KIX was expressed with a His tag and first purified using nickel sepharose affinity chromatography as described above. TAZ2 was expressed as an MBP fusion

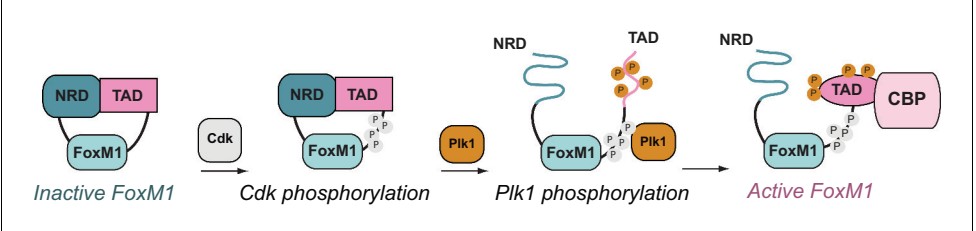

**Figure 7.** FoxM1 activation through conformational switching of the transactivation domain (TAD). (Left) Autoinhibited conformation, in which the TAD binds the negative regulatory domain (NRD) as a β-hairpin. (Center) Cyclin-dependent kinase (Cdk) phosphorylation (gray) creates a docking site for the Plk1 kinase, which phosphorylates the TAD and induces TAD release from the NRD. Both the NRD and TAD become structurally disordered upon dissociation. (Right) The phosphorylated TAD adopts a helical fold and recruits the coactivator CBP.
DOI: https://doi.org/10.7554/eLife.46131.018

protein and purified using amylose sepharose affinity chromatography and S-sepharose ion exchange chromatography. The TAZ2 contained cysteine to alanine point mutations (C1776A, C1784A, C1827A, C1828A) that increase solubility and stability (*De Guzman et al., 2000*). $ZnSO_4$ was kept in the media for TAZ2 expression at a concentration of 1 μM. TAZ1 was expressed with a His-Nus-XL affinity tag and purified using nickel affinity resin followed by tag cleavage. All CBP domains were further purified with Superdex 75 size exclusion chromatography.

## Isothermal titration calorimetry

Equilibrium dissociation constants for FoxM1 NRD constructs binding to TAD constructs were obtained using ITC with a MicroCal VP-ITC system. FoxM1 protein fragments were run over Superdex-75 into 20 mM HEPES, 150 mM NaCl, 5% glycerol (pH 7.5). TAD fragments (100–500 μM) were titrated into the NRD fragments (10–50 μM) at 25℃. The equilibrium dissociation constant for FoxM1 CSR binding to the Plk1 polobox domain was similarly determined. CSR (300 μM), either phosphorylated or unphosphorylated with Cdk2-CycA, was titrated into polobox (30 μM) at 25℃. Reported $K_d$ values are the average fits from two or three technical replicates with the standard deviation reported as error. Values for $K_d$, n, and ΔH are reported in *Supplementary file 1*. For data fitting in experiments involving the FoxM1 NRD, we adjusted NRD concentration such that the stoichiometry was close to 1. We chose this approach for the fitting, because we otherwise noticed variability in the n value that we attribute to difficulty measuring the NRD concentration from its low extinction coefficient. If no significant heat was detected above background, we conclude no association. In the case of titrating the TAD fragment 694–726 into 1–203, we observed small heats but could not fit the data, so we conclude the affinity must be weaker than 10 μM.

Equilibrium dissociation constants for CBP domain binding to FoxM1 TAD were obtained using ITC with Micro Cal VP-ITC system. Both FoxM1 and CBP proteins were run over Superdex-75 into 20 mM HEPES, 150 mM NaCl, 5% glycerol v/v (pH 7.5). CBP-TAZ2 (250–350 μM) was titrated into FoxM1 TAD (20–40 μM) at 25℃. Because of the difficulty in determining CBP-TAZ2 concentrations, the concentrations were adjusted to a stoichiometry of 1.0. CBP-KIX (600 μM) was titrated into FoxM1 TAD (60 μM) at 30 ℃ in the same buffer. Reported $K_d$ values are the average fits from two or three technical replicates with the standard deviation reported as error.

## Kinase reactions

FoxM1 protein constructs following final purification were incubated with 10 mM ATP, 50 mM $MgCl_2$, and 10% by mass of either Plk1 kinase domain, Cdk2-CycA, or both Plk1 and Cdk2-CycA, overnight at 4℃. Phosphorylation of the protein was confirmed by electrospray mass spectrometry using a Sciex X500B QTOF system.

## Circular dichroism

Circular dichroism spectra of the FoxM1 proteins, prepared in 20 mM sodium phosphate and 50 mM NaCl buffer (pH 7.0), were measured on a JASCO J-1500 spectrometer using a 1 mm path length quartz cuvette. Data were collected every 0.1 nm from 190 to 300 nm using a scanning speed

of 50 nm/min, a digital integration time of 4 s, and a bandwidth of 4 nm. Eight sets of data were collected for each protein. Protein concentrations ranged from 35 μM to 10 μM and were measured using absorption at 280 nm using a nanodrop spectrophotometer.

## Protein expression for NMR

Zebrafish FoxM1 (DrFoxM1) transactivation domain (G571-K623) and NRD-TAD fusion (*Figure 2— figure supplement 1A*) were expressed and purified from *E. coli* as above except the cells were grown in M9 medium supplemented with $^{15}$N ammonium chloride and $^{13}$C glucose. For isoleucine, leucine, and valine specific isotope labeling the cells were grown as above in deuterated M9 medium supplemented with deuterated $^{12}$C glucose and the specific precursors added 30 min before induction as described (*Tugarinov and Kay, 2003*). For the isotopomer-selective TOCSY experiment, DrFoxM1 was grown in $D_2O$ M9-media using $^{13}$C glucose ($^1$H,$^{13}$C-glucose). For the ILV sample, residues L19, L104, C87, L569, V571, L622 were mutated to alanine to reduce the number of overlapped peaks in the methyl spectra and to prevent disulfide formation (*Figure 2—figure supplement 1A*). For each NMR sample, the protein was purified as above and cleavage of the TEV linker was monitored by SDS-PAGE.

## Nuclear magnetic resonance data collection and assignments

### FoxM1 NRD-TAD fusion

NMR experiments for resonance assignment were performed on a Bruker Avance III HD 800-MHz spectrometer equipped with a cryogenically cooled probe head. All samples were prepared in a buffer containing 20 mM sodium phosphate pH 6.3, 100 mM KCl, and 5% (v/v) $D_2O$. The NMR spectra for backbone resonances assignments were collected using a 270 μM uniformly $^{13}$C, $^{15}$N–labeled deuterated DrFoxM1 NRD-TAD. Sequence-specific backbone resonance assignments for DrFoxM1 NRD-TAD were determined using all TROSY-HSQC, TROSY-HNCO, TROSY-HN(CA)CO, TROSY-HNCA, TROSY-HN(CO)CA, TROSY-HNCB, and TROSY-HN(COCA)CB experiments supplied by Bruker BioSpin. NMR spectra for methyl sidechain assignments were collected using 270 μM deuterated uniformly $^{15}$N-labeled, methyl site specific $^{13}$C-labeled protein. Methyl sidechain assignments were determined using an isotopomer-selective TOCSY, SOFAST NOESY, HMQC, and methyl-HSQC experiments (*Otten et al., 2010*; *Rossi et al., 2016*). Data from the protein samples with methyl specific labels were collected using SOFAST pulse sequences (*Rossi et al., 2016*), including amide to amide NOESY ($H_N$-$NH_N$ and N-$NH_N$), amide to methyl ($H_NH_{Aro}$-$C_MH_M$ and $C_M$-$NH_N$) and methyl to methyl ($H_M$-$C_MH_M$ and $C_M$-$C_MH_M$) SOFAST NOESY experiments all recorded with a recycle delay of 0.2 s and an NOE mixing time of 300 msec.

### FoxM1 TAD

NMR experiments for resonance assignment were performed on a Bruker Avance III HD 800-MHz spectrometer equipped with a cryogenically cooled probe head. All samples were prepared in a buffer containing 20 mM Sodium Phosphate pH 6.3, 100 mM KCl, and 5% (v/v) $D_2O$. The NMR spectra for backbone resonances assignments were collected using a 450 μM uniformly $^{13}$C, $^{15}$N–labeled DrFoxM1 TAD protein. NMR labeled TAD protein was phosphorylated by Plk1 kinase domain as described above. Sequence-specific backbone resonance assignments for DrFoxM1 TAD were determined using HSQC, HNCO, HNCACB, CBCA(CO)NH, and C(CO)NH experiments supplied by Bruker BioSpin. Sequence-specific backbone resonance assignments for phosphorylated DrFoxM1 TAD were determined using HSQC, HNCO, HNCACB, and CBCA(CO)NH experiments supplied by Bruker BioSpin.

### Residual dipolar coupling

The backbone $D_{NH}$ (NH) dipolar couplings used for RASREC-Rosetta structure modeling were measured using uniformly $^{15}$N-labeled deuterated FoxM1 NRD-TAD protein. The NMR sample was made by mixing DrFoxM1 NRD-TAD (250 μM final concentration) with Pf1 phage (12.5 mg/mL final concentration) and 10% $D_2O$ (*Hansen et al., 1998*). RDCs were collected on a Varian INOVA 600 MHz spectrometer using J modulation experiments similar to those described (*Tjandra et al., 1996*). The $D_{NH}$ coupling experiment was performed with the ILV NRD-TAD sample containing the alanine mutations (*Figure 2—figure supplement 1A*).

A separate set of backbone $D_{NC'}$ (NCO) dipolar couplings were measured using uniformly $^{15}N/^{13}C/^{2}D$ labeled FoxM1 NRD-TAD fusion protein and used for structure validation. This set of RDCs was not used in the structural calculation of the FoxM1 model. The NMR sample was made by mixing DrFoxM1 NRD-TAD (250 μM final concentration) with Pf1 phage (12.5 mg/mL final concentration) and 10% $D_2O$ (*Hansen et al., 1998*). RDCs were collected on a Varian INOVA 600 MHz spectrometer using J modulation experiments similar to those described previously (*Liu and Prestegard, 2010*). The $D_{NC'}$ coupling experiment was performed with the uniformly $^{15}N/^{2}D$-labeled NRD-TAD fusion sample (no alanine mutations) used for backbone assignments.

All NMR data were processed with NMRPipe and NMRDraw (*Delaglio et al., 1995*). Chemical-shift assignments were made with SPARKY (https://www.cgl.ucsf.edu/home/sparky/).

## Structure modeling using RASREC-Rosetta

Structural models of the NRD-TAD complex were calculated using RASREC-Rosetta (*Nerli and Sgourakis, 2019*). RASREC-Rosetta is a Monte Carlo-based fragment assembly approach that utilizes NMR chemical shifts to guide the conformation search for a near-native structure. Together with NMR chemical shifts, which aid in the selection of accurate secondary structural elements, we utilized NOE and RDC measurements that helped identify the correct fold of the two domains (*Nerli et al., 2018*). Alongside NMR data, RASREC-Rosetta uses optimized algorithms across six stages of resampling in a parallelized manner to achieve high structural convergence. During the initial stages of the protocol, various β-sheet topologies are sampled. In the subsequent stages, fragments derived from (i) high-resolution X-ray structures, and (ii) preliminary low-resolution conformations (from the initial sampling stages), are applied to intensify and finalize the folds of a target protein. In the final stages of the protocol, the low-resolution models generated during the initial and intermediate stages are refined in the Rosetta force field to produce high-resolution structures.

Structure calculations were set up with automated Python scripts available at the CS-Rosetta website (https://csrosetta.chemistry.ucsc.edu). Prior to setting up these calculations, we (i) removed flexible end regions in the target sequence based on TALOS-N RCI-$S^2$ predictions, and (ii) used the trimmed target sequence and chemical shift values to pick structural fragments of amino acid lengths 3 and 9 (standard lengths). After these steps, we used the protein sequence, structural fragments, chemical shift, NOE and RDC measurements as input to the RASREC-Rosetta protocol. From a set of 100 models generated by RASREC-Rosetta in its final stage, we selected 30 lowest energy models from which we filtered ten converged structures that fit the experimental RDC data. We subsequently refined the final ensemble twice (first, using NOE and RDC data sets and second, without any data sets to eliminate heavy bias from NMR data) using Rosetta's relax protocol (*Tyka et al., 2011*). The structures of NRD-TAD domains were calculated by treating the protein as a single polypeptide chain with the TEV site as a linker, which was later removed from the models.

To calculate the structural models of the FoxM1 NRD-TAD domains, we used a total of 64 NOE distance restraints derived from highly sensitive SOFAST-based experiments recoded with short inter scan delays (200 msec) and long (300 msec) mixing times (*Supplementary file 2*). These distance restraints consist of (i) 14 amide to amide, (ii) 11 amide to methyl, and (iii) 39 methyl to methyl NOEs. The observed intensities of the NOEs can be affected by distance-independent processes such as T1 relaxation and spin diffusion during the long NOE mixing time. Accordingly, we considered several additional parameters, alongside the signal intensities of the NOEs (relative to their diagonals), to calibrate upper distance bounds for all the three classes of NOE restraints described above. In particular, we performed an analysis of distance statistics within β-sheet structures in the PDB, and used preliminary models of the NRD-TAD domains calculated using more generous (7 Å), uniform NOE distance limits as benchmarks. First, we filtered NOEs based on high signal intensities. From a non-redundant set of 5 β-sheet X-ray structures in PDB, we found that anti-parallel β-strands have NOEs between pairs of amide protons within ~3 Å or ~5 Å and parallel β-strands within ~3 Å or ~4 Å. We then used the approximate relation that NOE signal intensities are inversely proportional to sixth power of distances ($I_{NOE} = c \times r^{-6}$, where $I_{NOE}$ is the NOE signal intensity, $c$ is the proportionality constant and $r$ is the distance) to obtain estimates of upper distance bounds. To determine the proportionality constant, we used the strongest-intensity NOE and approximate distances between the amide protons from β-sheet X-ray structures in PDB. Further, we applied the estimated proportionality constant and measured signal intensities to generate upper distance bounds for

other amide-amide NOEs. Similarly, we used preliminary models computed using fixed distance bounds to calibrate distances of amide-methyl and methyl-methyl NOEs. Using this process, we observed that the resulting convergence of NRD-TAD ensemble calculated by RASREC-Rosetta increased progressively, together with the optimization of structural quality parameters such as Rosetta energies and MOLPROBITY scores.

## Cell culture

Human osteosarcoma U2OS cells were maintained in Dulbecco's modified Eagle's medium (DMEM) containing 10% fetal bovine serum, 2 mM L-glutamine, and 1x Penicillin Streptomycin and split every 2–3 days. Cells tested negative for Mycoplasma contamination using a PCR assay.

Reporter assays were performed using the FoxM1c isoform, which includes 15 amino acids downstream of the DNA binding domain that are absent in FoxM1b. The amino acid sequences of the NRD, CSR and TAD domains are identical in FoxM1c and FoxM1b. For clarity, mutations are indicated using FoxM1b numbering throughout the figures and text. Mutants were expressed from the pCDNA3 plasmid with an N-terminal FLAG tag: pCDNA3-FOXM1C (WT), pCDNA3-FOXM1C-I62A I64A, pCDNA3-FOXM1C-V76A I78A, pCDNA3-FOXM1C-I88A, pCDNA3-FOXM1C-I88A L91A T92A, pCDNA3-FOXM1C-F106A L108A, pCDNA3-FOXM1C-V723A (V708A in FOXM1b), pCDNA3-FOXM1C-L724A (L709A in FOXM1b), pCDNA3-FOXM1C-V723A L724A (V708A L709A in FOXM1b), pCDNA3-FOXM1C-L731A (L716A in FOXM1b), pCDNA3-FOXM1C-S730A (S715A in FOXM1b). In co-expression experiments with Plk1, Plk1 was expressed from pRcCMV myc-Plk1 wt (a gift from Erich Nigg) (*Golsteyn et al., 1994*). The day prior to transient transfections, $2 \times 10^5$ U2OS cells were seeded into 12-well plates in antibiotic-free medium (DMEM/10% FBS/L-glutamine). Cells were then transfected with Lipofectamine 2000 Reagent (Invitrogen 11668–019) according to the manufacturer's protocol with 650 ng pGL3-6DB-reporter or pGL3-PLK1 promoter reporters (a kind gift from Piotr Sicinski, *Anders et al., 2011*). 650 ng pcDNA3-FOXM1C (WT or mutant), and 10 ng of pCMV-Renilla luciferase. In experiments that assayed activation by Plk1, 160 ng of pRcCMV myc-Plk1 wt or an empty vector control was also included. Media was refreshed 4 hr post-transfection. Cell lysates were prepared 25 hr post-transfection using the Promega Dual Luciferase Reporter Assay System kit (Promega, E1960) and luminescence was measured on a Promega Glo Max. Relative luminescence was determined by normalizing the Firefly luciferase activity (expressed by pGL3 reporter plasmids) activity to Renilla Luciferase (transfection efficiency control). Three technical replicate transfections were performed in each experiment and each experiment was carried out three times (biological replicates). Analysis was performed in Excel and GraphPad Prism software.

## Western blot and cell cycle analysis

The day prior to transient transfections, $2 \times 10^6$ U2OS cells were seeded into 10 cm plates in antibiotic-free medium (DMEM/10% FBS/L-glutamine). Cells were then transfected with Lipofectamine 2000 Reagent (Invitrogen 11668–019) according to the manufacturer's protocol with 9.5 μg pcDNA3-FOXM1 plasmids. Media was refreshed 4 hr post-transfection. Cell lysates were prepared 25 hr post-transfection by resuspending cells in RIPA buffer (150 mM NaCl, 1% Triton-X 100, 0.5% Sodium deoxycholate, 0.1% SDS, 50 mM Tris-Cl pH 8.0, 5 mM Sodium Fluoride, 1 mM Sodium Orthovanadate, 80 mM β-glycerophosphate, 1 μg/mL Leupeptin, 1 μg/mL bestatin, 1 μM Benzamidine HCl, 1 mM DTT) for 30 min on ice, followed by centrifugation. Protein concentration was determined using Bio-Rad Protein Assay Dye Reagent (Cat#5000006) and measured on an Eppendorf BioPhotometer Plus. Lysates were added to 4X SDS-PAGE Sample Buffer (0.25 M Tris pH 6.8, 8% SDS, 40% glycerol, 20% β-mercaptoethanol) and heated to 95℃ for 5 min. 10–20 μg total protein was then subjected to SDS–polyacrylamide gel electrophoresis (SDS–PAGE), followed by transfer to nitrocellulose membranes, and Western blotting with the following antibodies: rabbit anti-FoxM1 (Bethyl A301-533A-M), mouse anti-actin (Sigma A1978), mouse anti-Flag (Sigma F1804). For cell cycle analysis, cells were fixed in 70% ethanol at 4 ℃ overnight. Cells were then centrifuged, washed with PBS, washed with PBS with 0.5% Tween20, then incubated with IFA 5 μg/ml RNaseA for 30 min at 37 ℃. Propidium iodide was added to a final concentration of 50 μg/ml. Data were collected using a Guava EasyCyte HT (Millipore) and analyzed with FlowJo software.

## Data availability

NMR backbone assignments for zebrafish FoxM1 TAD and phosphorylated TAD are available from the BMRB with accession number 27763 and 27764 respectively. Coordinates, NOE and RDC restraint lists of NRD-TAD domains are available under PDB accession number 6OSW with corresponding chemical shift measurements available from BMRB accession number 30608.

## Acknowledgements

This work was supported by grants from the National Institutes of Health to JAB (R01GM117152), NGS (R35GM125034, R01AI143997), and SMR (R01GM127707, R01GM124148) and grants to SMR from the American Cancer Society (RSG-12-131-01-CCG) and Alex's Lemonade Stand Foundation. The JASCO J-1500 CD spectrophotometer was acquired with an NIH Shared Instrumentation Grant (S10OD016246), and the 800 MHz NMR spectrometer at UCSC was funded by the Office of the Director, NIH, under High End Instrumentation (HIE) Grant S10OD018455. We thank Miguel Osorio for optimization of the CBP-TAZ2 purification.

## Additional information

### Funding

| Funder | Grant reference number | Author |
| --- | --- | --- |
| National Institute of General Medical Sciences | R01GM117152 | Jennifer A Benanti |
| National Institute of General Medical Sciences | R35GM125034 | Nikolaos G Sgourakis |
| National Institute of Allergy and Infectious Diseases | R01AI143997 | Nikolaos G Sgourakis |
| NIH Office of the Director | S10OD018455 | Nikolaos G Sgourakis Seth M Rubin |
| American Cancer Society | RSG-12-131-01-CCG | Seth M Rubin |
| Alex's Lemonade Stand Foundation for Childhood Cancer | | Seth M Rubin |
| National Institute of General Medical Sciences | R01GM127707 | Seth M Rubin |
| National Institute of General Medical Sciences | R01GM124148 | Seth M Rubin |

The funders had no role in study design, data collection and interpretation, or the decision to submit the work for publication.

### Author contributions

Aimee H Marceau, Caileen M Brison, Conceptualization, Investigation, Methodology, Writing—original draft; Santrupti Nerli, Investigation, Methodology, Writing—original draft; Heather E Arsenault, Investigation, Data acquisition and analysis; Andrew C McShan, Investigation, Data analysis; Eefei Chen, Hsiau-Wei Lee, Investigation, Methodology; Jennifer A Benanti, Nikolaos G Sgourakis, Conceptualization, Supervision, Investigation, Methodology, Writing—original draft; Seth M Rubin, Conceptualization, Supervision, Funding acquisition, Writing—original draft

### Author ORCIDs

Andrew C McShan http://orcid.org/0000-0002-3212-9867
Jennifer A Benanti http://orcid.org/0000-0003-2484-5721
Nikolaos G Sgourakis http://orcid.org/0000-0003-3655-3902
Seth M Rubin https://orcid.org/0000-0002-1670-4147

**Decision letter and Author response**
Decision letter https://doi.org/10.7554/eLife.46131.031
Author response https://doi.org/10.7554/eLife.46131.032

## Additional files

### Supplementary files

• Supplementary file 1. Summary of ITC data fitting.
DOI: https://doi.org/10.7554/eLife.46131.019

• Supplementary file 2. NMR restraints and structural statistics for NRD-TAD structural ensemble.
DOI: https://doi.org/10.7554/eLife.46131.020

• Transparent reporting form
DOI: https://doi.org/10.7554/eLife.46131.021

### Data availability

Nuclear Magnetic Resonance Data have been deposited in the BMRB under accession number 27763, 27764, and 30608. Structural coordinates have been deposited in the PSB under accession number 6OSW.

The following datasets were generated:

| Author(s) | Year | Dataset title | Dataset URL | Database and Identifier |
|---|---|---|---|---|
| Marceau AH, Caileen M Brison, Santrupti Nerli, Andrew C McShan, Hsiau-Wei Lee, Nikolaos G Sgourakis, Seth M Rubin | 2019 | An order-to-disorder structural switch activates the FoxM1 transcription factor | http://www.rcsb.org/structure/6OSW | Protein Data Bank, 6OSW |
| Marceau AH, Caileen M Brison, Santrupti Nerli, Andrew C McShan, Hsiau-Wei Lee, Nikolaos G Sgourakis, Rubin SM | 2019 | An order-to-disorder structural switch activates the FoxM1 transcription factor | http://www.bmrb.wisc.edu/data_library/summary/?bmrbId=30608 | Biological Magnetic Resonance Data Bank, 30608 |
| Marceau AH, Brison CM, Nerli S, Arsenault HE, McShan AC, Chen E, Lee H-W, Benanti JA, Sgourakis NG, Rubin SM | 2019 | FoxM1 Transactivation Domain | http://www.bmrb.wisc.edu/data_library/summary/?bmrbId=27763 | Biological Magnetic Resonance Data Bank, 27763 |
| Marceau AH, Brison CM, Nerli S, Arsenault HE, McShan AC, Chen E, Lee H-W, Benanti JA, Sgourakis NG, Rubin SM | 2019 | FoxM1 Transactivation Domain, Phosphorylated form | http://www.bmrb.wisc.edu/data_library/summary/?bmrbId=27764 | Biological Magnetic Resonance Data Bank, 27764 |

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
