## [Decision Letter]

Thank you for submitting your article "An order-to-disorder structural switch activates the FoxM1 transcription factor" for consideration by *eLife*. Your article has been reviewed by three peer reviewers, including Lewis E Kay as the Reviewing Editor and Reviewer #1, and the evaluation has been overseen by Cynthia Wolberger as the Senior Editor.

The reviewers have discussed the reviews with one another and the Reviewing Editor has drafted this decision to help you prepare a revised submission.

As you can see the reviewers all agreed that the work is of high quality and general interest.

Marceau et al. reveal how the accessibility of a transactivation domain (TAD) that is predicted to be intrinsically disordered can be blocked by a negative regulatory domain (NRD), and the different roles that phosphorylation can play in regulating TAD access. These domains are contained in FoxM1, a critical activator of mitotic gene expression normally expressed in dividing cells, and a target for therapeutic intervention due to its role in multiple cancers. Solution NMR and CD spectroscopy showed that both the TAD and NRD undergo a disorder-to order transition to interact in a globular structure. Phosphorylation at a single site in the TAD by Plk1 kinase was sufficient to disrupt this interaction giving rise to a model where FoxM1 activation requires phosphorylation by Cdk to create a binding site for Plk1, which then phosphorylates the TAD, disrupting its interaction with NRD. This model is well supported by binding affinities measured by ITC, and FoxM1 transactivation assays in osteosarcoma cells. Overall, a detailed understanding of the inhibition of FoxM1 transactivation by NRD, and the role of phosphorylation in this process is well supported by the data. The authors took advantage of appropriate labeling and structure calculation strategies to use the limited amount of data that could be obtained from this challenging system to determine structures of sufficient quality to reveal these insights. Overall, the paper is clearly written, the figures are well-presented, and the data is of good quality, with analysis of interactions relevant for both FoxM1 activation and its transactivation activity. The work is novel for the binding-induced structure revealed in 2 intrinsically disordered domains for the activation of a key transcriptional activator in mitosis.

1) There is a minor issue with the CD data (Figure 5) that clearly shows a structural difference between the NRD-TAD fusion and isolated TAD and NRD, consistent with the proposed binding-coupled folding. Specifically, the shape of the spectra for isolated domains seem to reflect significant secondary structure content, which would be interesting to compare to the fusion by secondary structure deconvolution. In addition, there is a statement that the CD spectrum of the NRD-TAD fusion shows both α helical and β structure, which is difficult to see from the data shown, but would be better supported if the secondary structure deconvolution were done. This would require that protein concentrations be accurately determined. Some description of how protein concentrations were measured for spectra would be useful to assess the accuracy of the intensities shown. (The signal intensities are surprisingly different between these different constructs, as well as the reference spectra shown in Figure 5—figure supplement 1A.)

2) The figure legend for 2 (A) does not indicate the reason for boxed residues in the TAD sequence alignment.

3) Is it possible, through the addition of large excess amounts of co-activator domains to shift the equilibrium towards the active state. That is, while phosphorylation is clearly required for the physiological response, in a test-tube can large amounts of co-factor overcome the thermodynamic barrier? Or does phosphorylation (in addition to leading to unfolding) also increase binding in another way? Is there any evidence of a skewed equilibrium between inactive (NRD-TAD bound) and a minor state involving an activated complex? Have the authors tried relaxation dispersion experiments to test for this?

4) There is a concern in the use of the NRD-TAD fusion construct to solve the structure of the interdomain interaction. While the fusion construct is a better mimic of the native modular architecture of the protein, the authors should indicate potential caveats with using the fusion. For example, as demonstrated here, and in many instances in the literature, IDPs/IDRs' are very plastic, with seemingly little changes (i.e. solution conditions, PTMs, binding partner, protein expression tags) having significant effects on their conformation state. To make sure that the presence and/or the nature of the linker is not what is inducing this structure it is suggest that the authors think of a way of comparing the spectra (NMR, CD, Fluorescence, DSC thermogram or any technique that's sensitive to structure) of the apo NRD, apo TAD, complex NRD + TAD (i.e. trans) with that of the fusion NRD-TAD construct (i.e. cis). Can the authors show data of the labeled isolated NRD (TAD) titrated with unlabeled isolated TAD (NRD) and show that spectra obtained overlay with that of the NRD-TAD fusion. This should demonstrate the presence of the co-disorder-to-order transition upon binding.

5) The authors should discuss the enrichment of IDRs in transcription factors to show that these proteins are truly modular with all (or most) consisting of folded domains intersperse with IDRs. The following review could be of interest (Babu et al., 2012).

6) The results in this manuscript should be discussed in the context of other published PTM-mediate disorder-to-order (Pufall et al., 2005; Bah et al., 2015) and/or order-to-disorder transitions (Mitrea et al., 2014), to demonstrate the universality of IDR-folding plasticity for mediating biological function.

7) The authors may wish to perform bioinformatics disorder prediction of the NRD and TAD to complement the NMR/CD experiments. The program IUPRED will do (https://iupred2a.elte.hu/plot).

8) The authors should provide a table that shows all the thermodynamic parameters derived from the ITC data (i.e. the stoichiometry, enthalpy and entropy changes) in addition to the KDs.

9) It's difficult to follow which constructs are used for which assay and why. Sometimes it's the two minimal (or longer) constructs being tested in one assay, but at other times, it's one longer and one minimal (in the section of "Disruption of the NRD-TAD interaction activates FoxM1"). In the Reporter assays, FoxM1c isoform was used.

10) In the Discussion, the authors state "Our observation of specificity in the effects of FoxM1 phosphorylation more resembles what has been proposed for the Cdk inhibitor Sic1". I don't think this statement is accurate enough. In the Sic1 story, multiple phospho-motifs bind to the same site on the Cdc4 using the same phospho-motifs themselves, resulting in a dynamic complex.

---

## [Author Response]

1) There is a minor issue with the CD data (Figure 5) that clearly shows a structural difference between the NRD-TAD fusion and isolated TAD and NRD, consistent with the proposed binding-coupled folding. Specifically, the shape of the spectra for isolated domains seem to reflect significant secondary structure content, which would be interesting to compare to the fusion by secondary structure deconvolution. In addition, there is a statement that the CD spectrum of the NRD-TAD fusion shows both α helical and β structure, which is difficult to see from the data shown, but would be better supported if the secondary structure deconvolution were done. This would require that protein concentrations be accurately determined. Some description of how protein concentrations were measured for spectra would be useful to assess the accuracy of the intensities shown. (The signal intensities are surprisingly different between these different constructs, as well as the reference spectra shown in Figure 5—figure supplement figure 1A.)

We attempted secondary structure deconvolution as suggested using the BeStSel web server (Nucleic Acids Res. 2018 Jul 2;46(W1):W315-W322). In agreement with the reviewers’ expectations, we found foremost that both the quality of the fit and the output secondary structure composition were highly sensitive to small changes in input concentration. We note in the experimental section that we measure concentrations using UV 280 nM absorption, but we have concerns of the accuracy of NRD and NRDTAD fusion concentrations because of the low extinction coefficient of the NRD. We also learned that performance of the deconvolution alogrithim is hampered by the noise in our data below 200 nM. For these reasons, we are hesitant to quantify composition and prefer to keep our interpretation focused on comparing qualitatively the degree to which the domains appear structured or not. The analysis did agree with the reviewer observation that the shapes of the isolated TAD and NRD domain spectra reflect the presence of some secondary structure, and we now note this point in the relevant Results section:

“In contrast, the TAD and NRD spectra indicate relatively greater disorder, although they do reflect some secondary structure content.”

Because we feel we cannot accurately perform the secondary structure deconvolution, we removed the comment that the fusion spectrum indicates both α and β structure.

2) The figure legend for 2 (A) does not indicate the reason for boxed residues in the TAD sequence alignment.

The boxed residues are the Plk1 phosphorylation sites. We now indicate this reason in the caption.

3) Is it possible, through the addition of large excess amounts of co-activator domains to shift the equilibrium towards the active state. That is, while phosphorylation is clearly required for the physiological response, in a test-tube can large amounts of co-factor overcome the thermodynamic barrier? Or does phosphorylation (in addition to leading to unfolding) also increase binding in another way? Is there any evidence of a skewed equilibrium between inactive (NRD-TAD bound) and a minor state involving an activated complex? Have the authors tried relaxation dispersion experiments to test for this?

We performed an NMR experiment in which we titrated unlabeled TAZ2 into 100 µM 15N-labeled NRDTAD complex (the same cleaved fusion construct for which the structure was determined). We did observe perturbations in the spectrum upon addition of 500 µM TAZ2 (Figure 6—figure supplement 1), which suggest weak affinity (we estimate Kd ~0.1-1 mM). As the reviewers propose, we conclude in the figure caption that even in the unphosphorylated, “inactive” state, there is likely an equilibrium that is modulated by the activting phosphorylation:

“The observation that TAZ2 binds at high concentration indicates that the TAD is in an equilibrium between a sequestered and accessible state even when FoxM1 is unphosphorylated and that phosphorylation shifts the equilibrium.”

4) There is a concern in the use of the NRD-TAD fusion construct to solve the structure of the interdomain interaction. While the fusion construct is a better mimic of the native modular architecture of the protein, the authors should indicate potential caveats with using the fusion. For example, as demonstrated here, and in many instances in the literature, IDPs/IDRs' are very plastic, with seemingly little changes (i.e. solution conditions, PTMs, binding partner, protein expression tags) having significant effects on their conformation state. To make sure that the presence and/or the nature of the linker is not what is inducing this structure it is suggest that the authors think of a way of comparing the spectra (NMR, CD, Fluorescence, DSC thermogram or any technique that's sensitive to structure) of the apo NRD, apo TAD, complex NRD + TAD (i.e. trans) with that of the fusion NRD-TAD construct (i.e. cis). Can the authors show data of the labeled isolated NRD (TAD) titrated with unlabeled isolated TAD (NRD) and show that spectra obtained overlay with that of the NRD-TAD fusion. This should demonstrate the presence of the co-disorder-to-order transition upon binding.

We have added as Figure 2—figure supplement 2 an overlay of the HSQC spectra of the zebrafish fusion construct with the linker cleaved and uncleaved. While the quality of the uncleaved fusion is relatively poor with some peaks missing, it is clear that the dispersion and pattern of the peaks are similar, and we feel confident that the linker does not drastically perturb the overall structure. We agree that we cannot be certain that the structural model presented here is exactly the same as in the full-length protein (a problem often faced by structural biologists!); however, we emphasize that the structural model strongly correlated with the mutagenesis results from the functional assay in cells.

We realize in light of the reviewers’ comments that we were not clear enough in the manuscript that the structure was solved with the linker cleaved, and we have tried to make this more explicit in the main text and in the new supplemental figure.

We have faced a technical challenge preventing us from doing the titration experiments as suggested. We cannot express soluble zebrafish NRD on its own. We can for the human protein, allowing us to do the in trans ITC binding experiments, but the human protein does not produce NMR data of sufficient quality. It is for this reason that the final zebrafish construct for structure determination was expressed as a fusion and then cleaved.

5) The authors should discuss the enrichment of IDRs in transcription factors to show that these proteins are truly modular with all (or most) consisting of folded domains intersperse with IDRs. The following review could be of interest (Babu et al., 2012).

We have added the following sentences to the Introduction and cited the Babu et al. review.

“Structural characterization of transcription factors remains challenging as they are enriched in low complexity or intrinsically disordered domains (Babu et al., 2011). Transcription factors may have folded domains, such as a DNA binding domain (DBD), but these domains are typically modular and interspersed among disordered sequences.”

6) The results in this manuscript should be discussed in the context of other published PTM-mediate disorder-to-order (Pufall et al., 2005; Bah et al., 2015) and/or order-to-disorder transitions (Mitrea et al., 2014), to demonstrate the universality of IDR-folding plasticity for mediating biological function.

We have added the following text to the Discussion in order to discuss this previous work and the importance of IDP plasticity and regulation.

“Phosphorylation has been observed to control structural transitions as a means to regulate other intrinsically disordered proteins. For example, both the transcription factor Ets^-1^ and the translation regulator 4E-BP2 undergo disorder-to-order transitions upon phosphorylation, while the nucleophosmin protein dissociates from a structured pentamer to a disordered monomer upon phosphorylation (Bah et al., 2015; Mitrea et al., 2014; Pufall et al., 2005). FoxM1 demonstrates a new example in which the order-to-disorder transition occurs coincident with modulation of an intramolecular association between domains. This plasticity of disordered domains and the ability to regulate the plasticity through posttranslational modifications explains why intrinsically disordered proteins are so well suited to regulate biological function.”

7) The authors may wish to perform bioinformatics disorder prediction of the NRD and TAD to complement the NMR/CD experiments. The program IUPRED will do (https://iupred2a.elte.hu/plot).

We performed IUPRED analysis as suggested. IUPRED predicts disorder in the NRD but predicts the TAD to be rather ordered. This resuls is at odds with the experimental NMR data (Figure 5C and 5D), which demonstrates the TAD alone is predominantly disordered. It is likely the structural propensity reflected in the TAD sequence is present because it forms structure upon binding the NRD or coactivator. We have chosen not to include the IUPRED analysis, as we find the experimental data more convincing.

8) The authors should provide a table that shows all the thermodynamic parameters derived from the ITC data (i.e. the stoichiometry, enthalpy and entropy changes) in addition to the KDs.

We have included values for Kd, n (stoichiometry), and ΔH for each ITC experiment in an additional supplementary table (now Supplementary file 1). We choose not to report ΔS. While entropy could be calculated from the fit parameters for Kd and ΔH, we do not interpret ΔS or ΔH values to draw any conclusions in the manuscript, and we do not want to imply any interpretation. We include ΔH, because it is a fit parameter in determining the binding affinity from ITC data.

9) It's difficult to follow which constructs are used for which assay and why. Sometimes it's the two minimal (or longer) constructs being tested in one assay, but at other times, it's one longer and one minimal (in the section of "Disruption of the NRD-TAD interaction activates FoxM1"). In the Reporter assays, FoxM1c isoform was used.

We have done our best to clarify where possible in the text. For the TAD, we use the short construct (696-748) in all assays except when fusions are made to the CSR region to look at effects of Cdk phosphorylation. For the NRD, we used the short construct (1-114) in all structural assays, e.g. CD and NMR, and the long construct (1-203) in all ITC binding assays.

We explained this construct choice for ITC assays in the “Disruption of the NRD-TAD interaction…” section:

“For these experiments, we used the TAD (696-748) and a longer NRD (1-203), which were the most stable constructs containing each domain in solution and the simplest to purify.”

We added the following rationale for construct choice in the CD experiments:

“We used the minimal NRD construct (1-114) for CD analysis, because sequence analysis and NMR data suggest the region between 115-203 is disordered.”

10) In the Discussion, the authors state "Our observation of specificity in the effects of FoxM1 phosphorylation more resembles what has been proposed for the Cdk inhibitor Sic1". I don't think this statement is accurate enough. In the Sic1 story, multiple phospho-motifs bind to the same site on the Cdc4 using the same phospho-motifs themselves, resulting in a dynamic complex.

We agree that the Sic1 story is complex, as Sic1 phosphorylation has elements of both specificity and redundancy. In order to avoid confusion, we have removed Sic1 as an example.